# UNIFIED GENERATIVE MODELING FOR MULTIMODAL TIME SERIES ANALYSIS

## ABSTRACT

Modeling multimodal time series has become an emerging research focus, aiming to incorporate auxiliary modalities, such as textual descriptions, into time series analysis. This integration enables a deeper understanding of temporal patterns by leveraging diverse sources of information. However, existing approaches often treat external modalities merely as supplementary domain features, neglecting the joint distribution between time series and auxiliary modalities. Moreover, most prior methods are tailored to specific tasks, limiting their generality and the effective utilization of multimodal data. In this paper, we propose GENTS, a unified generative model for multimodal time series analysis, integrating a variety of downstream tasks within a unified modeling framework. GENTS is trained to generate time series from textual descriptions and to forecast future values conditioned on historical multimodal data, simultaneously. This approach enables the model to capture the joint distribution between time series and external modalities, supporting a broad range of applications such as conditional generation, forecasting, and time series editing. Furthermore, by incorporating time series captioning as an integral component, GENTS has largely alleviated the common challenge of multimodal data scarcity. Extensive experiments on diverse real-world datasets demonstrate the effectiveness and generality of our approach across multiple tasks.

## 1 INTRODUCTION

Time series are prevalent across real-world applications, including energy (Lai et al., 2018), climate (Jing et al., 2024b; 2021), healthcare (He et al., 2023; Chen et al., 2024; Jarrett et al., 2021), and finance (Gao et al., 2024). However, they rarely appear in isolation and are often accompanied by auxiliary modalities. For instance, a patient's physiological signals may be paired with diagnostic reports (Wagner et al., 2020), or climate indicators may be supplemented by expert summaries (Xu et al., 2024a). These multimodal signals not only enrich time series data and provide complementary context for interpretation, but also help uncover the underlying factors driving time series generation, enabling more effective analysis. As a result, a growing body of research focuses on integrating auxiliary modalities to enhance time series analysis.

Multimodal time series analysis explores leveraging external modality in various downstream tasks. Forecasting is most widely studied (Liu et al., 2024a;b), where textual descriptions are commonly used to improve prediction accuracy. Another direction addresses time series generation (Lee et al., 2023; Narasimhan et al., 2024), conditioning the generation process on meta-information such as categories or attributes. Beyond generation, time series editing (Jing et al., 2024a) seeks to modify existing time series in response to newly provided information.

While prior studies have demonstrated the effectiveness of incorporating multimodal data for time series analysis, the way to leverage the external modality information with consideration of time series remains an open problem. Existing works (Li et al., 2025; Liu et al., 2024a;b) often treat external modalities merely as supplementary features, such as domain-specific metadata, without modeling the joint distribution between time series and auxiliary modalities. Moreover, these approaches typically rely on task-specific architectures due to differences in modeling design and data formats, limiting their generalizability and the full potential of multimodal time series data.

To address the above limitations, we propose GENTS, a unified generative framework that captures the joint distribution of time series and auxiliary modalities, specifically, textual descriptions[1]. We use the term *unified* in a precise sense: it denotes a single backbone and training pipeline that support diverse time-series tasks, rather than separate task-specific models or repeated retraining. We introduce a novel perspective by framing diverse time series tasks within a unified generative paradigm. In this view, each

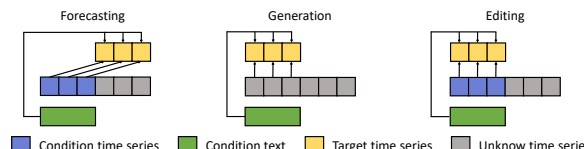

Figure 1: The target time series and multimodal conditions in different tasks (forecasting, generation, and editing). Forecasting is to generate future time series conditioned on historical ones and textual description, while generation (editing) aims at producing a time series based on the given textual information (and an original time series for modification).

task involves modeling the distribution of target time series given specific conditions, which may include time series segments, textual descriptions, or both, as explained in Fig. 1. GENTS is trained to generate a target segment of the time series conditioned on multimodal inputs, enabling seamless adaptation to a variety of downstream tasks using an individual model. This unified approach not only streamlines solutions across tasks but also enhances data efficiency and overall performance.

In addition, multimodal modeling requires large amounts of paired multimodal data, yet high-quality pairs like time series with textual descriptions are scarce. Early attempts used coarse domain knowledge (Jin et al.; Niu et al., 2025), while others relied on online reports (Liu et al., 2024b) or expert annotations (Guo et al., 2022), which are either weakly correlated or costly to scale. Recent captioning methods also depend on domain-specific, high-quality text for training. To overcome this, we propose a domain-agnostic captioning approach that integrates knowledge from both the historical time series and external tools, which provides additional paired multimodal information when the expert knowledge is absent.

Our contributions are concluded as follows: (i) We introduce GENTS, a novel generative modeling method that models joint distribution of multimodal time series data, and unifies multiple tasks of time series analysis into a whole framework, which is effective and adaptable in various downstream tasks. (ii) Our method incorporates an automated time series captioning method, which has significantly alleviated the scarcity issue of multimodal time series data and eventually improves modeling. (iii) We constructed a comprehensive benchmark for generative modeling on various time series tasks, and proved the efficiency of the unified framework compared with the task-specific methods.

## 2 RELATED WORK

### 2.1 MULTIMODAL TIME SERIES ANALYSIS

Many studies (Liu et al., 2024b; Zhang et al., 2025) have shown that numerical values in time series data alone provide limited information, while incorporating additional multimodal data offers valuable complementary insights. Various modalities have been explored to enhance time series analysis. Time-VLM (Zhong et al., 2025) transforms time series into images and leverages the pretrained vision-language models (VLMs). Frequency has also been considered as an alternative perspective for interpreting time series, as ContextTST (Hong et al., 2025) models the frequency to enhance model generalization. Among all modalities, text stands out as the most intuitive for humans to understand time series and is widely adopted in multimodal time series modeling. In our work, we take text modality as external information and explore multimodal time series analysis.

Previous literature has studied incorporating texts into time series modeling. In forecasting, several works introduce domain-level descriptions (Niu et al., 2025; Liu et al., 2024c) or sample-specific reports retrieved from the Internet (Liu et al., 2024b; Xu et al., 2024b) as supplementary inputs to improve forecasting. However, such text often lacks both high relevance and semantic alignment with the corresponding time series. To address this limitation, some studies (Trabelsi et al., 2025; Li et al., 2023; Lee et al., 2025) attempt to interpret time series using human language, typically relying on high-quality, paired datasets of text and time series to train captioning models. Nevertheless, these methods tend to be dataset-specific and lack generalizability across domains. In time series

---

[1]Without generality, we focus on textual information as the external modality for time series analysis.

generation, several works utilize metadata to guide the generation of new time series (Narasimhan et al., 2024; Li et al., 2022) or to modify existing ones (Jing et al., 2024a). Although metadata—often a simplified form of text—can be useful, it typically has a rigid structure and is insufficient for fine-grained control. Recent efforts (Jin et al.; Jia et al., 2024) aim to harness the capabilities of large language models (LLMs) by learning unified representations of both time series and text. These advancements motivate us to further explore the joint modeling of these two modalities.

## 2.2 TIME SERIES MODELING

Time series modeling can be generally categorized into discriminative and generative modeling, each serving different purposes depending on the task requirement. In forecasting, discriminative modeling has been the dominant approach. Early deep learning methods (Nie et al., 2022; Zeng et al., 2023) and foundation models (Ansari et al., 2024) treat forecasting as a supervised regression problem, directly mapping historical sequences to future numerical values. Recent advancements have introduced generative modeling as an alternative framework for forecasting (Yuan & Qiao; Su et al., 2025). Generative models learn the probability distribution of future sequences conditioned on history, enabling uncertainty estimation and diverse forecasts, crucial for high-variance and risk-sensitive domains like finance, weather, and healthcare. In generation and editing, generative modeling has become the standard paradigm. These tasks typically emphasize diversity, flexibility, and control over the generated outputs. Works include generating time series under specific constraints (Coletta et al., 2023), discrete metadata (Narasimhan et al., 2024; Li et al., 2022; Lee et al., 2023), and modifying existing sequences based on attributes (Jing et al., 2024a). Discriminative models are limited to single deterministic outputs, making them unsuitable for such tasks.

Addressing diverse time series tasks often requires the design of task-specific architectures and training objectives. For instance, forecasting, generation, and editing may each demand different model hypotheses, inductive biases, data representations, and optimization strategies. Recent work (Kollovieh et al., 2023) explores unified frameworks via unimodal unconditional generative pre-training. However, it struggles with conditional modeling, particularly in handling multimodal inputs and enabling fine-grained control, due to the absence of explicit conditioning mechanisms.

## 3 GENTS: UNIFIED GENERATIVE MODELING FOR TIME SERIES

In this work, we propose GENTS, a unified generative modeling framework designed to learn the joint distribution of time series and textual modality. We begin by formulating the multimodal generative modeling in Sec. 3.1 and introducing the backbone of our approach. Sec. 3.2 presents our unified modeling paradigm, followed by Sec. 3.3, where we describe how GENTS is adapted to a range of multimodal time series tasks. To address the challenge of limited paired data, we further propose a domain-agnostic time series captioning method in Sec. 3.4.

## 3.1 TIME SERIES GENERATIVE MODELING

Given pairs of time series $\mathbf{x}$ and multimodal conditions $\mathbf{c}$ drawn from the joint distribution $q(\mathbf{x}, \mathbf{c})$, our objective is to learn a generative model $p_\theta(\mathbf{x}, \mathbf{c})$ that closely approximates the true joint distribution $q(\mathbf{x}, \mathbf{c})$. With condition $\mathbf{c}$ sampled from the real distribution $q(\mathbf{c})$, we have $p_\theta(\mathbf{x}, \mathbf{c}) = p_\theta(\mathbf{x}|\mathbf{c})q(\mathbf{c})$. To this end, we adopt the diffusion model (Ho et al., 2020) as the backbone of our generative framework, owing to its proven stability and effectiveness in capturing complex data distributions, along with its acceptable inference efficiency, which together make it well-suited for unifying multiple time series tasks with multimodal data.

During the training phase of the diffusion model, the forward process gradually adds noise to the original data sample $\mathbf{x}_0 = \mathbf{x}$ through a Gaussian Markov transition[2]:

$$q(\mathbf{x}_{1:T}|\mathbf{x}_0) = \prod_{t=1}^{T} q(\mathbf{x}_t|\mathbf{x}_{t-1}), \quad q(\mathbf{x}_t|\mathbf{x}_{t-1}) = \mathcal{N}(\mathbf{x}_{t-1}; \sqrt{1-\beta_t}\mathbf{x}_{t-1}, \beta_t\mathbf{I}), \quad (1)$$

where $\{\beta_t\}_{t=1}^{T}$ are the predetermined variance schedule. Sampling of $\mathbf{x}_t$ has the closed-form written as $\mathbf{x}_t = \sqrt{\alpha_t}\mathbf{x}_0 + \sqrt{1-\alpha_t}\boldsymbol{\epsilon}$, where $\boldsymbol{\epsilon} \sim \mathcal{N}(\mathbf{0}, \mathbf{I})$ and $\alpha_t := \prod_{s=1}^{t}(1-\beta_s)$. To recover the original

---

[2] $\mathbf{x}$ and $\mathbf{x}_0$ are interchangeable in this paper.

data $\mathbf{x}_0$ from the noisy version $\mathbf{x}_t$, a conditional noise estimator $\epsilon_\theta(\mathbf{x}_t, t, \mathbf{c})$ is trained to estimate the noise added to $\mathbf{x}_0$. The objective function can be written as:

$$\min_\theta \mathcal{L}(\mathbf{x}_0) = \min_\theta \mathbb{E}_{\boldsymbol{\epsilon} \sim \mathcal{N}(\mathbf{0},\mathbf{I}), t \sim \mathcal{U}(1,T)} \| \boldsymbol{\epsilon} - \epsilon_\theta(\mathbf{x}_t, t, \mathbf{c}) \|_2^2. \tag{2}$$

During the inference phase, given a condition $\mathbf{c}$, the sample $\hat{\mathbf{x}}_0$ can be generated from a gaussian noise $\mathbf{x}_T \sim \mathcal{N}(0, \mathbf{I})$ through multiple denoising steps, which utilize the noise estimator $\epsilon_\theta(\mathbf{x}_t, t, \mathbf{c})$ with the deterministic Denoising Diffusion Implicit Model (DDIM) (Song et al., 2020) sampler:

$$\hat{\mathbf{x}}_{t-1} = \frac{\sqrt{\alpha_{t-1}}}{\sqrt{\alpha_t}}(\hat{\mathbf{x}}_t - \sqrt{1 - \alpha_t}\epsilon_\theta(\hat{\mathbf{x}}_t, t, \mathbf{c})) + \sqrt{1 - \alpha_{t-1}}\epsilon_\theta(\hat{\mathbf{x}}_t, t, \mathbf{c}). \tag{3}$$

The diffusion model has been widely adopted in time series modeling due to its stable and powerful distribution modeling capabilities (Ho et al., 2020). However, its application has largely been limited to single-task scenarios, such as imputation (Tashiro et al., 2021), forecasting (Yuan & Qiao), generation (Narasimhan et al., 2024), and editing (Jing et al., 2024a). We argue that the full potential of diffusion models in capturing complex data distributions remains underexplored. Therefore, in the following sections, we investigate how diffusion models can be leveraged for universal generative modeling, aiming to construct a unified generative framework supporting multiple time series tasks.

## 3.2 Unified Generative Modeling

We formulate various time series tasks as *masked-conditional generation* problems and develop a unified generative framework based on diffusion models (Ho et al., 2020). The central challenge in building such a unified model lies in standardizing the input–output interface across tasks, while allowing the model to distinguish among task-specific settings.

**Masked-Conditional Generation.** We introduce some notations below to support our formulation:

- $\mathbf{x} \in \mathbb{R}^L$: the original time series sequence of length $L$.
- $\mathbf{c}_{\text{text}} \in \mathbb{N}^W$: the textual description of the time series, where $W$ is the number of tokens.
- $\mathbf{m}_{\text{cond}} \in \{0,1\}^L$: a binary mask indicating conditional elements of $\mathbf{x}$. A value of one marks elements used as conditions while others keep zero.
- $\mathbf{m}_{\text{tgt}} \in \{0,1\}^L$: a binary mask indicating target elements to be generated.

As illustrated in Fig. 1, the data formats vary across different tasks. Specifically, forecasting generates future time series conditioned on historical observations and textual descriptions, while generation and editing aim to produce or modify a time series based on textual information (and an original series for editing). Despite their differences, these tasks share a common structure: chronologically aligned condition and target segments of the time series, accompanied by corresponding textual descriptions. Thus, as shown in Fig. 2, we separately regard the time series $\mathbf{x}$ as $\mathbf{x}_{\text{cond}}$ and $\mathbf{x}_{\text{tgt}}$ using the condition mask $\mathbf{m}_{\text{cond}}$ and target mask $\mathbf{m}_{\text{tgt}}$, respectively.

Under our unified generative paradigm, we instantiate the time series $\mathbf{x}$ and multimodal condition $\mathbf{c}$ from the general diffusion formulation (Sec. 3.1) as follows: The target $\mathbf{x}_{\text{tgt}}$ represents the portion to be generated; and the multimodal condition is defined as $\{\mathbf{x}_{\text{cond}}, \mathbf{c}_{\text{text}}\}$. Accordingly, the masked-conditional generation task is formulated as learning the conditional distribution $p_\theta(\mathbf{x}_{\text{tgt}} \mid \mathbf{x}_{\text{cond}}, \mathbf{c}_{\text{text}})$. By explicitly utilizing condition and target masks, our proposed framework flexibly distinguishes between conditioning inputs and generation targets across a wide range of tasks, enabling a unified approach to modeling the joint distribution of multimodal time series data.

**Unified I/O.** The input $\mathbf{x}_t$ of the unified diffusion model for training is constructed as follows:

$$\mathbf{x}_t = \mathbf{x}_{\text{cond}} + \mathbf{x}_{\text{tgt}} = \mathbf{m}_{\text{cond}} \circ \mathbf{x}_0 + \mathbf{m}_{\text{tgt}} \circ \left[ \sqrt{\alpha_t}\mathbf{x}_0 + \sqrt{1 - \alpha_t}\boldsymbol{\epsilon} \right], \tag{4}$$

where $\alpha_t$ and $\boldsymbol{\epsilon}$ are defined in Sec. 3.1. Specifically, $\mathbf{x}_t$ consists of three parts: (i) the condition segment corresponding to the index set: $\{i | \mathbf{m}_{\text{cond}}^i = 1\}$; (ii) the target segment for generation corresponding to the index set: $\{i | \mathbf{m}_{\text{tgt}}^i = 1\}$; (iii) the unknown segment corresponding to the index set: $\{i | \mathbf{m}_{\text{cond}}^i = 0, \mathbf{m}_{\text{tgt}}^i = 0\}$. In general, the input to the unified generative model includes the task type $\tau$, the time series sequence $\mathbf{x}_t$, the textual description $\mathbf{c}$, and the diffusion step $t$. The output is the estimated noise $\hat{\boldsymbol{\epsilon}}_t \in \mathbb{R}^L$, where $\hat{\boldsymbol{\epsilon}}_t = \epsilon_\theta(\mathbf{x}_t, \mathbf{c}_{\text{text}}, \tau, t)$.

**Unified Optimization Objective.** Our optimization objectives are primarily divided into two categories: forecasting the future and generating the history. Within the diffusion model framework, both tasks are uniformly formulated as the estimation of noise in the target generation segment, specified by the mask $\mathbf{m}_{\text{tgt}}$. Different masks are employed to distinguish between the task types:

$$\mathcal{L} = \sum_{\tau \in \{F,G\}} \mathbb{E}_{\boldsymbol{\epsilon} \sim \mathcal{N}(\mathbf{0},\mathbf{I}), t \sim \mathcal{U}(1,T)} \lambda_\tau \times ||\mathbf{m}_{\text{tgt}} \circ [\epsilon_\theta(\mathbf{x}_t, \mathbf{c}_{\text{text}}, \tau, t) - \boldsymbol{\epsilon}]||_2^2, \tag{5}$$

where $\circ$ is Hadamard product, $F$ and $G$ denote forecasting and generation, respectively, and $\lambda_\tau$ is a hyperparameter used to balance the loss contribution from each task. We will further discuss the variations in the condition mask $\mathbf{m}_{\text{cond}}$ and the target generation mask $\mathbf{m}_{\text{tgt}}$ in Sec. 3.3.

**Unified Model Architecture.** We design our model architecture with reference to (Peebles & Xie, 2023), which effectively models interactions between different modalities. The inputs are first transformed into embeddings through a series of encoders. As shown in Fig. 2, the noisy time series sequence $\mathbf{x}_t$ and the generation target mask $\mathbf{m}_{\text{tgt}}$ are encoded into the patch sequence $\mathbf{P} \in \mathbb{R}^{N \times D}$ and the patch mask embedding $\mathbf{m}_p \in \mathbb{R}^{N \times D}$, respectively, where $N = \lfloor \frac{L-P}{P} \rfloor$, $P$ is the patch size. The diffusion step $t$ is embedded into $\mathbf{e}_t \in \mathbb{R}^D$. To distinguish different tasks, we prepend the task type token $\tau$ to the textual condition $\mathbf{c}_{\text{text}}$ and use a text encoder to obtain the embedding $\mathbf{e}_c \in \mathbb{R}^D$. We model the influence of the time series condition and the textual condition through two mechanisms: self-attention and layer normalization adapter. The unified noise estimator consists of $J$ layers, each containing an adapter, a self-attention block, and a feed-forward network. For the $j$-th layer, the computation is defined as:

$$\beta, \gamma = \text{MLP}(\mathbf{e}_c),$$
$$\tilde{\mathbf{H}}^j = \mathbf{H}^j \times \beta + \gamma, \tag{6}$$
$$\mathbf{H}^j = \text{FFN}(\text{SA}(\tilde{\mathbf{H}}^{j-1})) + \mathbf{H}^{j-1},$$

where $\mathbf{H}^0 = \mathbf{P} + \mathbf{m}_p + \mathbf{e}_t$, MLP denotes the adapter module, SA is the self-attention block, and FFN is the feed-forward network. The interaction between the textual condition $\mathbf{c}_{\text{text}}$ and the noisy time series $\mathbf{x}_t$ is captured via the adapter MLP, while the interaction between the condition segment and target segment in $\mathbf{x}_t$ is modeled through the self-attention block SA. Finally, the output from the last layer $\mathbf{H}^J$ is passed through a patch decoder to produce the estimated noise $\hat{\epsilon}_t$. For more details of the unified noise estimator, please refer to Appendix D.

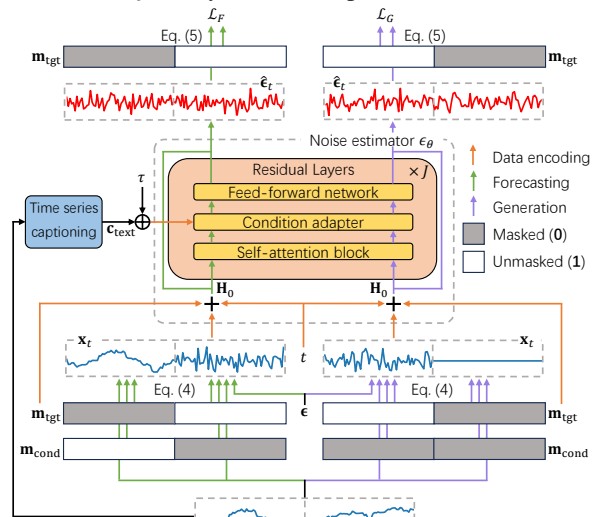

Figure 2: The overall pipeline of GENTS, including the unified generative modeling framework and domain-agnostic captioning.

Compared to the existing unimodal approach (Kollovieh et al., 2023), which unifies different tasks through unconditional generation combined with self-guidance, we introduce a unified generative modeling framework grounded in the masked-conditional generation paradigm to better address the challenges inherent in multimodal time series tasks. By leveraging the condition adapter and masked self-attention mechanism, our method enables more flexible processing of multimodal information, thereby improving its applicability across a broader range of scenarios.

## 3.3 TACKLING MULTIPLE TASKS OF TIME SERIES ANALYSIS

Following the unified generative modeling framework, GENTS addresses a variety of time series tasks under a common paradigm, to generate time series conditioned on multimodal inputs. As outlined in Sec. 3.2, different combinations of condition and target masks are employed to distinguish between task types within this unified formulation. To support this, we partition each time series sequence of length $L$ into two segments: a history segment of length $L_h$ and a future segment of length $L_f$, such that $L = L_h + L_f$. This segmentation provides a consistent structure across tasks, enabling flexible condition–target configurations based on the specific task requirements.

Table 1: Summary of attributes extracted using `tsfresh` (Christ et al., 2018).

| Attribute Type | Attribute Name | Description |
|---|---|---|
| **Statistical** | Mean/Std/Min/Max | The mean, standard deviation, minimum and maximum value of the time series. |
| | Skewness/Kurtosis | Measures the asymmetry and sharpness of the time series value distribution. |
| | Frequency | Identifies the dominant periodicity in the frequency domain. |
| **Morphological** | Overall/Local Trend | Captures the trend slope of the entire time series and each segment. |
| | Number of Peaks | Counts the number of local maxima within each segment. |
| | Complexity | Quantifies the roughness of each segment. |

**Forecasting.** Multimodal time series forecasting aims to predict future values based on historical sequences and accompanying texts, while strictly avoiding the use of future information to prevent data leakage. We construct the condition mask as $\mathbf{m}_{\text{cond}} = [1^{L_h}, 0^{L_f}]$ and the generation target mask as $\mathbf{m}_{\text{tgt}} = [0^{L_h}, 1^{L_f}]$. The input sequence is defined as $\mathbf{x}_t = \mathbf{m}_{\text{cond}} \circ \mathbf{x}_0 + \mathbf{m}_{\text{tgt}} \circ \hat{\mathbf{x}}_{t+1}$, where $\hat{\mathbf{x}}_T \sim \mathcal{N}(0, \mathbf{I})$, $T$ is the diffusion step number. A task-specific token "$< forecasting >$" is prepended to the textual condition $\mathbf{c}_{\text{text}}$ to indicate the forecasting task and provide contextual guidance.

**Generation.** Time series generation (Narasimhan et al., 2024) focuses on synthesizing a sequence solely based on a textual description. The condition mask is set as $\mathbf{m}_{\text{cond}} = \mathbf{0}^L$ and the generation target mask as $\mathbf{m}_{\text{tgt}} = [\mathbf{1}^{L_h}, \mathbf{0}^{L_f}]$. The input time series sequence is constructed as $\mathbf{x}_t = \mathbf{m}_{\text{cond}} \circ \mathbf{x}_0 + \mathbf{m}_{\text{tgt}} \circ \hat{\mathbf{x}}_{t+1}$, where $\hat{\mathbf{x}}_T \sim \mathcal{N}(0, \mathbf{I})$, $T$ is the diffusion step number. The token "$< generation >$" is prepended to the textual input $\mathbf{c}_{\text{text}}$ to indicate the task type.

**Editing.** Time series editing (Jing et al., 2024a) seeks to revise an existing sequence in accordance with new textual guidance. It uses the same mask configuration and task token "$< generation >$" as in the generation task. However, during inference, rather than initializing from Gaussian noise like generation task, the input is constructed by perturbing the original sequence through the forward diffusion process, following (Jing et al., 2024a): $\mathbf{x}_t = \mathbf{m}_{\text{cond}} \circ \mathbf{x}_0 + \mathbf{m}_{\text{tgt}} \circ \hat{\mathbf{x}}_{t+1}$, where $\hat{\mathbf{x}}_T = \sqrt{\alpha_T}\mathbf{x}_0 + \sqrt{1 - \alpha_T}\boldsymbol{\epsilon}, \boldsymbol{\epsilon} \sim \mathcal{N}(0, \mathbf{I})$, $T$ is the diffusion step number. The model then applies the learned noise estimator to denoise $\mathbf{x}_t$ conditioned on the updated textual input $\mathbf{c}_{\text{text}}$, yielding the edited output $\hat{\mathbf{x}}_0$.

### 3.4 DOMAIN-AGNOSTIC TIME SERIES CAPTIONING

To address the scarcity of paired time series and text, we propose a general framework for time series captioning inspired by how humans intuitively interpret unfamiliar time series: focusing on *statistical* and *morphological* attributes. Statistical attributes, such as mean, skewness, and kurtosis, summarize global distributional properties, while morphological attributes capture structural patterns like trends, peaks, and abrupt changes. Crucially, these attributes are domain-agnostic, offering a shared descriptive basis across diverse fields. For instance, both a temperature drop due to rainfall and a stock price decline from policy shifts can be described as a downward trend. This abstraction enables generalizable, informative captioning without relying on domain-specific knowledge.

In this section, we propose a general captioning method that extracts attributes from time series and converts them into textual descriptions using predefined templates. Our approach leverages tsfresh (Christ et al., 2018) as the attribute annotator, denoted by $\mathcal{T}$. We define a set of attributes to be extracted, with corresponding statistical and morphological parameters denoted by $\mathcal{P} = \{\mathcal{P}_{\text{sta}}, \mathcal{P}_{\text{mor}}\}$. The complete list of attributes considered is provided in Tab. 1. Given a time series $\mathbf{x}$, the attribute annotator $\mathcal{T}$ extracts the corresponding attribute set as follows:

$$\mathcal{A} = \mathcal{T}(\mathbf{x}, \{\mathcal{P}_{\text{sta}}, \mathcal{P}_{\text{mor}}\}), \tag{7}$$

where $\mathcal{A} = \{\mathcal{A}_i\}_{i=1}^M$ denotes the set of extracted attributes, and each $\mathcal{A}_i \in \mathbb{R}$ represents the value of the $i$-th attribute. These attribute values are discretized and converted into text using predefined templates. We provide more details about the text synthesis and quality evaluation in Appendix B.

Compared with existing time series captioning methods (Lee et al., 2025; Li et al., 2023; Trabelsi et al., 2025), which rely on training captioning models using high-quality text from specific domains, our approach enables cross-domain captioning without requiring training. Furthermore, our approach leverages both the knowledge from historical sequences and external tools, offering a more informative and effective solution for aligned text generation across diverse domains.

## 4 EXPERIMENT

In this section, we present the experiment settings and the corresponding results, organized around the following research questions (RQs): **RQ1**: Is unified generative modeling effective for multiple downstream tasks? **RQ2**: Does the synthetic domain-agnostic text improve the performance of different tasks? **RQ3**: What's the impact of the unified generative modeling in different tasks? The code will be published upon the acceptance of this paper.

### 4.1 EXPERIMENT SETUP

**Datasets.** We utilize two categories of datasets: (i) Multimodal datasets consisting of time series and manually collected textual descriptions. These include **TimeMMD-Env**, the largest subset of TimeMMD datasets (Liu et al., 2024b), **Weather** dataset (Xu et al., 2024a), and an original dataset **HongKong** (Hong Kong Observatory, 2025) collected by ourselves. (ii) Unimodal datasets consisting of time series only, including **ETTm1** (Zhou et al., 2021), **Exchange** (Lai et al., 2018), and **Traffic** (Leo, 2024). More details regarding dataset statistics and the specific data pairing protocol for the editing task are provided in Appendix C.

**Evaluation Metrics.** For the forecasting task, we report mean squared error (**MSE**) and mean absolute error (**MAE**). For the generation task, Joint Frechet Time Series Distance (**J-FTSD**) (Narasimhan et al., 2024) is used to quantify the discrepancy between the distribution of generated and real data; Contrastive Time Series Text Pretraining (**CTTP**) score is incorporated to measure the semantic similarity between the generated time series and their corresponding textual descriptions. For the editing task, we adopt the Log Ratio of Target-to-Source (**RaTS**) (Jing et al., 2024a) and the **CTTP** score to evaluate the model's ability to modify existing time series according to new textual descriptions. Further details on the evaluation metrics are provided in Appendix F.

**Baselines.** We compared our method with different baselines for generation, editing, and forecasting tasks. For generation and editing, the metadata-based method **TimeWeaver** (Narasimhan et al., 2024), **TEdit** (Jing et al., 2024a) are considered. For forecasting, we consider both the unimodal and multimodal methods. The unimodal models include the MLP method **Dlinear** (Zeng et al., 2023), CNN method **TimesNet** (Wu et al.), transformer method **Autoformer** (Wu et al., 2021), diffusion method **Diffusion-TS** (Yuan & Qiao, 2024), and foundation models **Moirai** (Woo et al., 2024) and **Chronos** (Ansari et al., 2024). The multimodal models include **TimeCMA** (Liu et al., 2024a) and **TimeMMD** (Liu et al., 2024b), **IATSF** (Xu et al., 2024b).

**Setting.** GENTS is implemented as a unified multi-task framework trained from scratch on each individual dataset, rather than a pre-trained foundation model. Specifically, for the main results reported in Tab. 2, a single GENTS model is trained jointly on all relevant tasks using only the training split of the target dataset. It is then evaluated on the corresponding test split of the same dataset. We adopt different configurations of history length, forecasting horizons, and generation length for different datasets. For the TimeMMD-Env, ETTm1, Exchange, and Traffic datasets, the setting is $(48, \{48, 96, 144\}, 48)$. The configuration is $(28, \{28, 56, 84\}, 28)$ for HongKong dataset and $(36, \{36, 72, 108\}, 36)$ for Weather dataset. On unimodal datasets, GENTS utilizes only the synthetic domain-agnostic text. On multimodal datasets, GENTS additionally incorporates the real textual descriptions provided in the datasets. GENTS is trained one round for different tasks, with a loss weight ratio of $\lambda_F : \lambda_G = 2 : 1$. We set the number of epochs to 700, the batch size to 1024, and the learning rate to $1 \times 10^{-3}$ across all datasets. All experiments were conducted on a single Nvidia-A40 GPU running three times with different random seeds.

### 4.2 QUANTITATIVE RESULTS

In this section, we quantitatively evaluate the generation, editing, and forecasting performance of GENTS and baselines across all unimodal and multimodal datasets. The results are presented in Tab. 2. Forecasting results are reported as averages across different horizons. Details for individual horizons forecasting, visual case studies demonstrating the editing and generation capabilities, and model efficiency are provided in Appendix G.

**Finding 1:** *A unified generative model can effectively handle generation, editing, and forecasting tasks, achieving strong performance across all of them.* As shown in Tab.2, GENTS delivers com-

Table 2: Averaged performance of generation, editing, and forecasting tasks on multimodal (left) and unimodal (right) datasets. The best results are shown in bold, and the second-best results are underlined. Arrows ↑ (↓) indicate that higher (lower) values are better.

| Task | Multimodal datasets | | | | | | Unimodal datasets | | | | | |
|---|---|---|---|---|---|---|---|---|---|---|---|---|
| **Generation** | TimeMMD-Env | | Weather | | HongKong | | ETTm1 | | Exchange | | Traffic | |
| | ↓JFTSD | ↑CTTP | ↓JFTSD | ↑CTTP | ↓JFTSD | ↑CTTP | ↓JFTSD | ↑CTTP | ↓JFTSD | ↑CTTP | ↓JFTSD | ↑CTTP |
| TimeWeaver (Narasimhan et al., 2024) | 421.145 | 79.728 | 291.369 | 55.809 | 458.184 | 120.509 | 40.281 | 93.554 | 78.282 | 85.068 | **140.031** | 55.520 |
| TEdit (Jing et al., 2024a) | 423.662 | 76.808 | 290.678 | 54.996 | 457.276 | 120.956 | 30.761 | 108.750 | 80.606 | 82.621 | 141.305 | 55.097 |
| GENTS (ours) | **166.379** | **83.687** | **1.320** | **115.251** | **23.516** | **140.052** | **27.227** | **127.575** | **45.801** | **180.377** | 178.937 | 44.942 |
| **Editing** | TimeMMD-Env | | Weather | | HongKong | | ETTm1 | | Exchange | | Traffic | |
| | ↑RaTS | ↑CTTP | ↑RaTS | ↑CTTP | ↑RaTS | ↑CTTP | ↑RaTS | ↑CTTP | ↑RaTS | ↑CTTP | ↑RaTS | ↑CTTP |
| TimeWeaver (Narasimhan et al., 2024) | 0.584 | 79.238 | 0.364 | 56.035 | 0.088 | 117.900 | 0.282 | 93.854 | −1.411 | 88.967 | −0.622 | 52.182 |
| TEdit (Jing et al., 2024a) | 0.556 | 76.011 | 0.370 | 55.962 | 0.078 | 117.916 | 0.551 | 107.843 | −1.577 | 83.941 | −0.498 | **56.721** |
| GENTS (ours) | **1.451** | **84.059** | **3.071** | **115.281** | **0.121** | **140.376** | **0.795** | **128.247** | **0.796** | **180.166** | **4.412** | 43.434 |
| **Forecasting** | TimeMMD-Env | | Weather | | HongKong | | ETTm1 | | Exchange | | Traffic | |
| | ↓MAE | ↓MSE | ↓MAE | ↓MSE | ↓MAE | ↓MSE | ↓MAE | ↓MSE | ↓MAE | ↓MSE | ↓MAE | ↓MSE |
| DLinear (Zeng et al., 2023) | 0.627 | 0.595 | 0.304 | 0.205 | 0.922 | 1.433 | **0.722** | 1.175 | 0.253 | 0.129 | 0.575 | 0.511 |
| Autoformer (Wu et al., 2021) | 0.835 | 1.218 | 0.444 | 0.377 | 1.833 | 5.505 | 1.257 | 2.815 | 0.949 | 1.795 | 2.942 | 12.350 |
| TimesNet (Wu et al.) | 0.538 | 0.556 | 0.288 | 0.207 | 0.786 | 1.108 | 0.790 | 1.380 | 0.183 | 0.066 | 0.486 | **0.381** |
| Diffusion-TS (Yuan & Qiao, 2024) | 0.532 | 0.587 | 0.421 | 0.406 | 0.820 | 1.233 | 0.746 | **1.107** | 0.323 | 0.198 | 0.575 | 0.610 |
| Moirai (Woo et al., 2024) | 0.736 | 1.629 | 0.349 | 0.280 | 0.888 | 1.458 | 1.422 | 1.751 | 0.180 | 0.069 | 1.157 | 2.109 |
| Chronos (Ansari et al., 2024) | 0.634 | 0.790 | 0.292 | 0.217 | 0.798 | 1.192 | 0.771 | 1.277 | 0.176 | **0.065** | 0.656 | 0.728 |
| TimeCMA (Liu et al., 2024a) | 0.534 | 0.538 | 0.361 | 0.270 | 0.780 | 1.072 | 0.791 | 1.339 | 0.185 | **0.065** | 0.926 | 1.072 |
| TimeMMD (Liu et al., 2024b) | **0.506** | **0.468** | 0.326 | 0.203 | 0.754 | 1.017 | 0.724 | 1.235 | 0.174 | 0.066 | 0.465 | 0.423 |
| IATSF (Xu et al., 2024b) | 0.510 | 0.514 | 0.267 | 0.169 | **0.735** | **0.974** | 0.767 | 1.245 | **0.166** | 0.066 | 0.916 | 1.036 |
| GENTS (ours) | 0.509 | 0.508 | **0.221** | **0.139** | 0.743 | 1.003 | **0.722** | 1.168 | 0.182 | **0.065** | **0.456** | 0.420 |

petitive performance on all three tasks. For generation and editing, GENTS consistently outperforms baselines on most datasets, benefiting from its ability to leverage fine-grained textual information. In forecasting, GENTS also demonstrates strong performance—achieving the best results or ranking second on almost all datasets, with scores closely matching the top-performing models. These results answer **RQ1**, confirming the effectiveness of the unified model across generation, editing, and forecasting tasks.

## 4.3 ABLATION STUDY

In this section, we present an ablation study of GENTS to examine the impact of unified modeling and the use of synthetic domain-agnostic text. Without loss of generality, we select the HongKong and Traffic datasets, representing a multimodal and a unimodal dataset, respectively.

**Finding 2:** *Synthetic domain-agnostic text improves forecasting accuracy and provides fine-grained conditions for generation.* As shown in Tab. 3, removing the textual descriptions leads to a significant performance drop in both forecasting and generation. For forecasting, the textual description offers an additional perspective on the historical sequence. For generation, it serves as a fine-grained condition that guides the generation process more precisely. This answers the **RQ2** that the synthetic domain-agnostic text benefits both tasks.

Table 3: The ablation study on HongKong and Traffic datasets. $\mathcal{L}_F$ is the forecasting loss, $\mathcal{L}_G$ is the generation loss, and $\mathbf{C}_{\text{text}}$ is the text condition. Since evaluating forecasting and generation without the corresponding losses $\mathcal{L}_F$ and $\mathcal{L}_G$ is meaningless, we leave these results blank.

| Method | Forecasting | | | | Generation | | | |
|---|---|---|---|---|---|---|---|---|
| | HongKong | | Traffic | | HongKong | | Traffic | |
| | ↓MAE | ↓MSE | ↓MAE | ↓MSE | ↓JFTSD | ↑CTTP | ↓JFTSD | ↑CTTP |
| GENTS | 0.743 | 1.003 | 0.456 | 0.420 | 23.516 | 140.052 | 178.937 | 44.942 |
| w/o $\mathcal{L}_F$ | - | - | - | - | 23.477 | 141.537 | 175.805 | 46.486 |
| w/o $\mathcal{L}_G$ | 0.764 | 1.065 | 0.563 | 0.601 | - | - | - | - |
| w/o $\mathbf{C}_{\text{text}}$ | 0.909 | 1.533 | 0.570 | 0.605 | 36.876 | 112.978 | 211.512 | 21.248 |

**Finding 3:** *Unified training of forecasting and generation can enhance forecasting performance while preserving generation quality.* As shown in Tab. 3, jointly modeling forecasting and generation yields better forecasting performance than modeling forecasting alone, while maintaining comparable generation quality. This suggests that both tasks can be formulated as conditional probabilistic modeling problems, and that their joint optimization enables forecasting to benefit from shared representations. These findings address **RQ3**, demonstrating that unified modeling not only preserves individual task performance but can also enhance the performance of specific tasks.

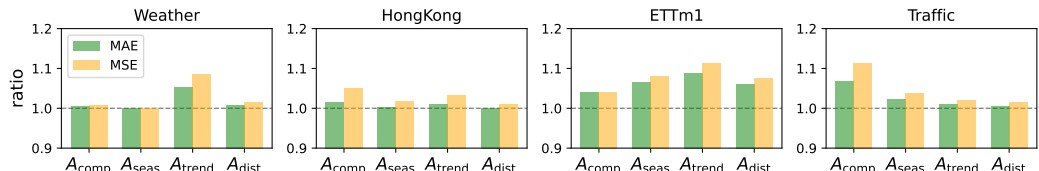

Figure 3: Error increase ratio of forecasting after *masking* the textual description of a certain class of attributes, where the higher ratio represents the larger impact on the forecasting performance. The masked attribute classes include $\mathcal{A}_{\mathrm{comp}}$ (complexity-related), $\mathcal{A}_{\mathrm{seas}}$ (season-related), $\mathcal{A}_{\mathrm{trend}}$ (trend-related), and $\mathcal{A}_{\mathrm{dist}}$ (distribution-related). The results are the average of different horizons on four datasets: Weather, HongKong, ETTm1, and Traffic.

## 4.4 EXTENDED ANALYSIS

**Finding 4:** *Statistical attributes are consistently beneficial for forecasting across all datasets, whereas morphological attributes contribute to varying degrees depending on the dataset.* To interpret the impact of different attributes on forecasting, we mask the corresponding textual descriptions and compute the error-increase ratios: $\frac{\mathrm{MAE_{mask}}}{\mathrm{MAE}}$ and $\frac{\mathrm{MSE_{mask}}}{\mathrm{MSE}}$. We categorize all the attributes (detailed in Tab. 1) into 5 classes: value-statistical (Mean/Std/Min/Max), complexity-related (Complexity/Number of Peaks), season-related (Frequency), trend-related (Overall/Local Linear Trend), and distribution-related (Skewness/Kurtosis). The results show that value-statistical attributes have a substantial impact across all datasets, with masking leading to an error increase ranging from 28% to 98%. In contrast, the influence of the other attributes varies across datasets. As shown in Fig. 3, each attribute class significantly affects the forecasting performance in at least one dataset, demonstrating that all extracted attributes are useful for forecasting.

**Finding 5:** *Synthetic domain-agnostic text can enhance zero-shot forecasting performance.* Distinct from the supervised evaluation in Tab. 2 where models are trained and tested on the same dataset, we conducted a separate analysis to probe the cross-dataset generalization capability of our domain-agnostic text features. As shown in Tab. 4, models augmented with domain-agnostic text achieve better performance when performing zero-shot forecast-

Table 4: The zero-shot forecasting performance of GENTS and GENTS without $\mathbf{C}_{\mathrm{text}}$. The model is trained on the dataset on the left of $\rightarrow$, and evaluated on the datasets on the right of $\rightarrow$.

| Method | Exchange→ Traffic | | ETTm1→ Traffic | | Exchange→ ETTm1 | |
| | ↓MAE | ↓ MSE | ↓MAE | ↓ MSE | ↓MAE | ↓ MSE |
| --- | --- | --- | --- | --- | --- | --- |
| GENTS | 1.206 | 2.230 | 1.034 | 1.515 | 0.985 | 2.112 |
| GENTS w/o $\mathbf{C}_{\mathrm{text}}$ | 1.404 | 3.192 | 1.060 | 1.694 | 1.050 | 2.401 |

ing on unseen datasets. This improvement indicates that domain-agnostic text captures the statistical and morphological characteristics of time series, which often reflect common physical patterns shared across different datasets.

**Finding 6:** *Task performances are robust to the hyperparameters, and similar settings generalize well across different datasets.* We conduct a sensitivity analysis on the loss weight ratio $\lambda_F : \lambda_G$. As shown in Fig. 4, except under extreme ratios, both forecasting and generation performances remain relatively stable, demonstrating the robustness of our method. Furthermore, we observe that different datasets exhibit similar preferences for the loss weight ratio. The most balanced configuration for the two tasks is around $\lambda_F : \lambda_G = 2 : 1$, which we adopt consistently across all datasets.

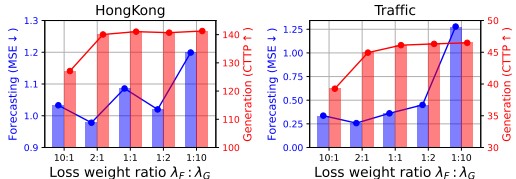

Figure 4: The sensitivity analysis of loss weight on HongKong (left) and Traffic (right) datasets. The loss weight ratio $\lambda_F : \lambda_G$ range in [10:1, 2:1, 1:1, 1:2, 1:10]. The further to the left, the higher the proportion of forecasting; otherwise, the higher the proportion of generation. MSE is reported for forecasting performance, CTTP is reported for generation performance.

## 5 CONCLUSION

The integration of multimodal information has become increasingly prominent in time series analysis. However, existing methods are often tailored to specific tasks, relying on specialized architectures and training strategies. Their scalability is further limited by the scarcity of large-scale

multimodal datasets. In this work, we introduce GENTS, a novel unified multimodal generative framework based on diffusion models that supports multiple time series tasks, including forecasting, generation and editing, within a unified architecture, enabling flexible adaptation and reducing the development cost and complexity. To mitigate the shortage of paired multimodal data, we further propose a domain-agnostic captioning method that automatically generates textual descriptions from statistical and morphological attributes of time series. Experiments demonstrate that GENTS consistently outperforms task-specific baselines across different tasks.

Despite its strengths, GENTS still has limitations. Currently, it only generates time series outputs. Nevertheless, we believe this work offers a promising step toward unified modeling of multimodal time series and can inspire future research on unified multimodal generative frameworks.

## 6 REPRODUCIBILITY STATEMENT

In this section, we summarize the information provided in the paper that facilitates reproducibility. Specifically, we introduce the unified generative framework GENTS for multi-task and multimodal modeling, with the overall design presented in Sec. 3.2 and a visualization shown in Fig. 2. The detailed architecture of GENTS is further described in Appendix D. For the proposed domain-agnostic captioning method, we present the implementation ideas in Sec. 3.4 and include the extracted attributes and text templates in Appendix B. In the experimental section, we report the configuration and hyperparameter settings in Sec. 4.1, provide detailed descriptions of data construction in Appendix C, and specify the evaluation metrics and the models they depend on in Appendix F and Appendix E. Overall, this paper offers comprehensive details spanning method design, data construction, and result evaluation, thereby ensuring reproducibility of the proposed work. We will release all the codes and datasets upon the acceptance of this paper.

## 7 ETHICS STATEMENT

This work adheres to the ICLR Code of Ethics. Our ethical considerations center on the responsible use of data and the societal impact of our models. All datasets utilized, such as TimeMMD-Env, HongKong, ETTm1, Exchange and Traffic, were sourced from public repositories and used in strict compliance with their terms of service. To ensure data privacy, the information is fully anonymized and contains no personally identifiable information (PII). Furthermore, we have integrated mitigation strategies throughout our research process to address potential algorithmic biases and prevent adverse societal outcomes, reaffirming our commitment to responsible research.

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

## A  THE USE OF LARGE LANGUAGE MODELS

In this work, large language models (LLMs) are primarily employed in data construction scenarios. As discussed in Appendix B.3, we use LLMs as an alternative approach to synthesize the textual descriptions of time series, treating the results as a baseline for evaluating the quality of the domain-agnostic text generated by our method.

## B  DOMAIN-AGNOSTIC TIME SERIES CAPTIONING

As mentioned in Sec. 3.4, we propose a domain-agnostic captioning method that first extracts a set of attributes $\mathcal{A}$ corresponding to the statistical and morphological parameters $\mathcal{P}_{\text{sta}}$ and $\mathcal{P}_{\text{mor}}$ from time series, and then converts them into textual descriptions using predefined templates. In this section, we provide a detailed explanation of the attribute extraction and text annotation process, which plays a key role in addressing the scarcity of multimodal data. Our pipeline comprises two main steps: (1) extracting attributes based on statistical and morphological parameters, and (2) converting these attributes into natural language using prompt templates. The entire process is fully automated and domain-agnostic.

### B.1  ATTRIBUTE LIST

For statistical attributes, we directly compute the *Mean*, *Standard Deviation*, *Minimum*, and *Maximum* of the time series data. We also use the `tsfresh` (Christ et al., 2018) library to extract *Skewness* and *Kurtosis* for measuring asymmetry and sharpness, as well as *Frequency* to identify the dominant periodicity in the frequency domain. For morphological attributes, we use the `tsfresh` (Christ et al., 2018) library to extract three types of features from the time series: *Overall/Local Trend*, *Number of Peaks*, and *Complexity*. A detailed list of the extracted attributes is presented in Tab. 1 in the main paper.

**Dataset-adaptive thresholds:** To ensure the interpretability of textual descriptions across diverse datasets, we define a parameter dictionary containing dataset-specific thresholds for trend slope and complexity (see Tab. 5). It is worth noting that these thresholds are derived by partitioning the empirical distribution of each attribute within a dataset into quantile intervals. The labels "high", "middle", and "low" correspond to different quantile ranges and are consistently used throughout the text mapping process.

Table 5: Dataset-specific thresholds for trend and complexity attribute mapping, set according to quantiles of the empirical distribution of each attribute within each dataset. $S_h$,$S_m$,$S_l$ denote high slope, middle slope, and low slope, respectively. $C_h$,$C_l$ denote high complexity and low complexity, respectively.

| **Dataset** | high slope $S_h$ | middle slope $S_m$ | low slope $S_l$ | high complexity $C_h$ | low complexity $C_l$ |
|---|---|---|---|---|---|
| TimeMMD-Env | 1.00 | 0.40 | 0.10 | 3.00 | 1.50 |
| HongKong | 1.00 | 0.40 | 0.10 | 2.00 | 0.50 |
| Weather | 0.30 | 0.10 | 0.01 | 0.14 | 0.01 |
| ETTm1 | 1.00 | 0.50 | 0.10 | 0.90 | 0.30 |
| Exchange | 0.30 | 0.10 | 0.02 | 0.17 | 0.05 |
| Traffic | 1.50 | 0.70 | 0.30 | 2.00 | 1.30 |

### B.2  TEXT CAPTIONING TEMPLATE

Following the attribute definition (see Tab. 1), we design text templates for two categories of attributes (3 types of attributes in the statistical category and 3 types of attributes in the morphological category).

#### B.2.1  TEXT TEMPLATE FOR STATISTICAL ATTRIBUTES

For statistical attributes, we design separate text templates for three categories of attributes: 1. *Mean/Std/Min/Max*, 2. *Skewness/Kurtosis*, and 3. *Frequency*.

*Mean/Std/Min/Max*. We directly calculate the mean $mean, standard deviation $std, minimum value $min, and maximum value $max of the time series. The text description is defined as:

- Mean: "The mean is {$mean}."
- Standard deviation: "The std is {$std}."
- Minimum: "The min value is {$min}."
- Maximum: "The max value is {$max}."

*Skewness*. The skewness of the time series distribution is calculated through `tsfresh.skewness` which measures the asymmetry of the value distribution. Based on the skewness value:

- If skewness $< -0.5$, the text description is: "The distribution of the value in time series is shifted to the negative."
- If skewness $> 0.5$, the text description is: "The distribution of the value in time series is shifted to the positive."
- Otherwise, the text description is: "The distribution of the value in time series is symmetrical."

*Kurtosis*. The kurtosis of the time series distribution is calculated through `tsfresh.kurtosis`, which measures the sharpness of the distribution. Based on the kurtosis value:

- If kurtosis $< -0.5$, the text description is: "and has low kurtosis."
- If kurtosis $> 0.5$, the text description is: "and has high kurtosis."
- Otherwise, the text description is: "and has normal kurtosis."

*Frequency*. The dominant frequency is identified via `tsfresh.fft_coefficient`. The frequency with the highest magnitude is described as: "The main season cycles is around {$n} pi", where {$n} is the index of the dominant frequency.

### B.2.2 TEXT TEMPLATE FOR MORPHOLOGICAL ATTRIBUTES

For morphological attributes, we design separate text templates for three categories of attributes: 1. *Overall/Local Trend*, 2. *Number of Peaks*, and 3. *Complexity*.

*Overall Trend*. The overall trend of the time series is calculated through `tsfresh.linear_trend`, which describes the trend direction and intensity. Based on the slope value $S$ and sequence length $L$ (with dataset-specific thresholds in Tab. 5), the textual description is randomly sampled from the following options:

For a positive slope ($S > 0$):

- Strong upward trend ($|S| > S_h/L$):
  - "The time series is going up rapidly."
  - "The time series has a sharp upward trend."
  - "The time series rises obviously."
- Medium upward trend ($S_m/L < |S| \leq S_h/L$):
  - "The trend direction is up."
  - "The time series is going upward."
  - "The time series increases by time."
- Weak upward trend ($S_l/L < |S| \leq S_m/L$):
  - "The time series slowly rises."
  - "The time series has a slow upward trend."
  - "The values in the time series are slightly climbing."
- Very weak upward trend ($|S| \leq S_l/L$):
  - "The time series has upward trend but not obvious."

For a negative slope ($S < 0$):

- Strong downward trend ($|S| > S_h/L$):
    - "The time series is going down rapidly."
    - "The time series has a sharp downward trend."
    - "The time series drops obviously."
- Medium downward trend ($S_m/L < |S| \leq S_h/L$):
    - "The trend direction is down."
    - "The time series is going downward."
    - "The time series decreases by time."
- Weak downward trend ($S_l/L < |S| \leq S_m/L$):
    - "The time series slowly drops."
    - "The time series has a slow downward trend."
    - "The values in the time series are slightly decreasing."
- Very weak downward trend ($|S| \leq S_l/L$):
    - "The time series has downward trend but not obvious."

*Local Trend.* For local trend, we first divide the time series into three segments: beginning, middle, and end along the time dimension equally. Then, the local trend for each segment is computed using `tsfresh.linear_trend`, with dataset-specific thresholds (see Tab. 5) for parameter settings. The description is randomly sampled from the following, according to the slope:

- For a strong upward trend: "At the $<$ segment $>$, the time series is going up rapidly."
- For a medium upward trend: "At the $<$ segment $>$, the time series is going upward."
- For a weak upward trend: "At the $<$ segment $>$, the time series slowly rises."
- For a very weak upward trend: "At the $<$ segment $>$, the time series has upward trend but not obvious."
- For a strong downward trend: "At the $<$ segment $>$, the time series is going down rapidly."
- For a medium downward trend: "At the $<$ segment $>$, the time series is going downward."
- For a weak downward trend: "At the $<$ segment $>$, the time series slowly drops."
- For a very weak downward trend: "At the $<$ segment $>$, the time series has downward trend but not obvious."

Where $<$ segment $> \in \{\text{beginning}, \text{middle}, \text{end}\}$.

*Number of Peaks.* For the number of peaks, similarly, the time series are first divided into three segments: beginning, middle, and end along the time dimension equally. Then, the number of local maxima in each segment is counted using `tsfresh.number_peaks`. If the number of peaks is greater than zero, the description would be in the following format:

- "At the beginning, there are 3 peaks."
- "At the middle, there are 2 peaks."
- "At the end, there are 1 peaks."

*Complexity.* For complexity, similarly, the time series are first divided into three segments: beginning, middle, and end along the time dimension equally. Then, the complexity of the pattern $C$ for each segment is measured using `tsfresh.cid_ce`. According to the dataset-specific thresholds (Tab. 5), the description is:

- If $C \leq C_l$: "the pattern complexity is low"
- If $C_l < C \leq C_h$: "the pattern complexity is middle"
- If $C_h < C$: "the pattern complexity is high"

### B.2.3 ATTRIBUTE INDEX

After extracting all features, each time series is also represented as a structured attribute index vector, which includes:

- **Variable Index:** $[0, K-1]$, where $K$ is the number of variables.
- **Trend Attribute:** $\{0, 1\}$, indicating $\{$upward, downward$\}$.
- **Seasonality Attribute:** $[0, 8]$, indicating the dominant periodicity.
- **Skewness Attribute:** $\{0, 1, 2\}$, indicating $\{$negative, positive, symmetrical$\}$.
- **Kurtosis Attribute:** $\{0, 1, 2\}$, indicating $\{$low, normal, high$\}$.

These attribute index vectors are also adopted as the structured representation of time series in attribute-based methods such as TimeWeaver (Narasimhan et al., 2024) and TEdit (Jing et al., 2024a).

### B.3 HUMAN EVALUATION ON TEXT QUALITY

To verify the quality of the synthesized domain-agnostic text, we conducted a human evaluation comparing our generated textual outputs with those from GPT-4, which was prompted to describe the same time series. Three researchers independently scored each text pair based on the corresponding time series figure, assigning 1 to the preferred and 0 to the other. We randomly sampled 100 pairs from the Weather dataset and report the average preference score. Our method was preferred over GPT-4. Qualitative analysis also showed that GPT-4's outputs contained irrelevant or hallucinated content, while ours were more focused and faithful. We will include this discussion in the revised version of the paper.

Table 6: The average human evaluation score between our domain-agnostic caption and the caption from GPT-4, the higher the better.

| Method | GENTS | GPT-4 |
|--------|-------|-------|
| Score↑ | 0.57 | 0.43 |

### B.4 ILLUSTRATIVE EXAMPLES

We provide an illustrative example of the generated caption in Figure 5. The left panel shows a univariate time series of Weather dataset, and the right panel displays the caption produced by our method. The caption jointly summarizes the series' global statistics, distributional shape, and morphological properties, resulting in a concise and human-readable description that is closely aligned with the underlying data.

## C DATASET DETAILS

In this section, we provide detailed information of the datasets, which include six datasets of two categories, as mentioned in Sec. 4.1.

### C.1 MULTIMODAL DATASETS

### C.1.1 TIMEMMD-ENV

The original dataset, TimeMMD-Env (Liu et al., 2024b), is a daily time series dataset focused on air quality monitoring in New York. The numerical data are sourced from the United States Environmental Protection Agency (EPA). This is a univariate dataset, with the core variable being the Air Quality Index (AQI)—a standardized measure of air pollution, where higher values indicate poorer air quality. The Environment subset spans the period from 1982 to 2023, encompassing over 40 years of historical data and totaling 11,102 timestamps.

In our experiments, we split the original dataset into training, validation, and test sets in an 8:1:1 ratio, resulting in 8,880 timestamps for the training set, 1,110 for the validation set, and 1,112 for the test set. To slice the long sequence into multiple samples, (history length, forecasting horizons, stride) is set as $(48, \{48, 96, 144\}, 1)$ following (Liu et al., 2024b).

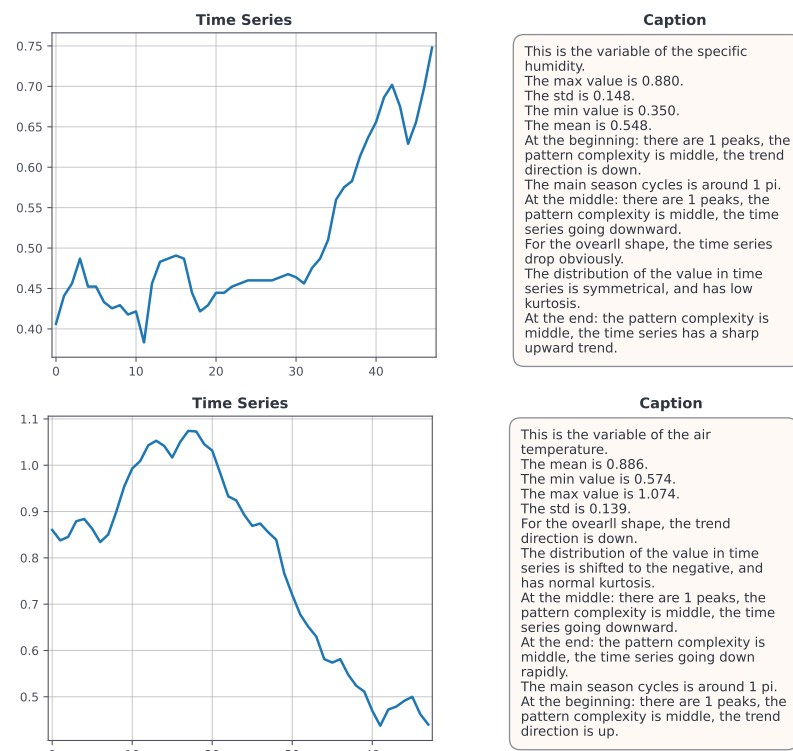

Figure 5: Illustrative examples of generated captions for univariate time series.

### C.1.2 WEATHER

Weather (Xu et al., 2024a) is a dataset of atmospheric signals collected by the Max Planck Institute for Biogeochemistry in Jena, Germany. It contains three variables: atmospheric pressure measured in millibars, temperature measured in degrees Celsius, and specific humidity measured in grams per kilogram. The original dataset spans from January 2014 to December 2023, with a sampling interval of 10 minutes.

In our experiments, we use data from 2014 to 2020 for training, 2021 for validation, and 2022 for testing, resulting in 366,912 timestamps for training, 52,560 for validation, and 52,128 for testing. The total timestamp number is 471,600 per variable. To slice the long sequence into multiple samples, (history length, forecasting horizon, stride) is set as $(36, \{36, 72, 108\}, 36)$.

### C.1.3 HONGKONG

HongKong is a daily dataset on weather conditions in Hong Kong, collected by us. We first gather raw daily data from March 2000 to April 2025, including numerical values and monthly notes from the official website of the Hong Kong Observatory (Hong Kong Observatory, 2025). We then extract numerical records from individual weather and monitoring stations, along with the corresponding monthly notes.

In our experiments, the HongKong dataset includes four variates: maximum air temperature, minimum air temperature, minimum relative humidity, and maximum relative humidity. We split the dataset into training, validation, and test sets using an 8:1:1 ratio, resulting in 7,237 samples for training, 906 for validation, and 904 for testing, totaling 9,047 samples for each variate. To slice the long sequence into multiple samples, we set the (history length, forecasting horizon) as $(28, 28, 56, 84)$. To ensure alignment with the corresponding textual descriptions, each historical sequence is constrained to fall within a natural calendar month.

## C.2 Unimodal Datasets

### C.2.1 ETTm1

ETTm1 (Zhou et al., 2021) is a dataset collected from an electricity transformer in China, with a sampling interval of 15 minutes. It contains seven variables: HUFL, HULL, MUFL, MULL, LUFL, LULL, and OT. The first six variates represent power load features, while OT refers to oil temperature. The dataset spans a period of two years.

In our experiments, we split the dataset into training, validation, and test sets using an 8:1:1 ratio, resulting in 55,744 samples for training, 6,968 for validation, and 6,968 for testing, totaling 69,680 samples for each variate. To slice the long sequence into multiple samples, (history length, forecasting horizon, stride) is set as (48, {48, 96, 144}, 48) following (Liu et al., 2024b).

### C.2.2 Exchange

Exchange (Lai et al., 2018) is a daily dataset containing the exchange rates of eight foreign countries: Australia, the United Kingdom, Canada, Switzerland, China, Japan, New Zealand, and Singapore. Each country corresponds to one variate, resulting in eight variates in total. The dataset spans from 1990 to 2016, with 7,588 daily samples for each variate.

In our experiments, we split the dataset into training, validation, and test sets using an 8:1:1 ratio, resulting in 6,064 samples for training, 766 for validation, and 758 for testing. To slice the long sequence into multiple samples, (history length, forecasting horizon, stride) is set as (48, {48, 96, 144}, 12) following (Liu et al., 2024b).

### C.2.3 Traffic

The original Traffic dataset (Leo, 2024) is a minutely time series dataset containing Traffic Index data for Istanbul, with three variables: Overall Traffic Index (TI), Asian Side Traffic Index (TI-An), and European Side Traffic Index (TI-Av). In our experiments, we resampled the original dataset at a ratio of 60, converting it into hourly data. The selected sampling period spans from November 1, 2022, to June 16, 2024, resulting in a total of 13,630 samples.

In our experiments, we split the dataset into training, validation, and test sets using an 8:1:1 ratio, resulting in 10,904 samples for training, 1,363 for validation, and 1,363 for testing. To slice the long sequence into multiple samples, (history length, forecasting horizon, stride) is set as (48, {48, 96, 144}, 4) following (Liu et al., 2024b).

## C.3 Data Construction for Time Series Editing Task

We follow the data construction process of TEdit (Jing et al., 2024a), but generalize from structured attributes to unstructured textual descriptions.

Let $\mathcal{D} = \{(\mathbf{x}_i, \mathbf{c}_i)\}_{i=1}^N$ denote a dataset of time——caption pairs, where $\mathbf{x}_i$ is a time series and $\mathbf{c}_i$ is its associated caption. We split $\mathcal{D}$ into training and test subsets and use only the test split $\mathcal{D}_{\text{test}} \subset \mathcal{D}$ for the editing experiments.

For each editing instance, we independently sample two distinct elements from $\mathcal{D}_{\text{test}}$:

$$(\mathbf{x}_{\text{src}}, \mathbf{c}_{\text{src}}), \ (\mathbf{x}_{\text{tgt}}, \mathbf{c}_{\text{tgt}}) \sim \mathcal{D}_{\text{test}}, \quad (\mathbf{x}_{\text{src}}, \mathbf{c}_{\text{src}}) \neq (\mathbf{x}_{\text{tgt}}, \mathbf{c}_{\text{tgt}}).$$

We refer to $\mathbf{x}_{\text{src}}$ as the original time series, and to $\mathbf{c}_{\text{tgt}}$ as the new textual guidance or editing instruction.

Given the input tuple $(\mathbf{x}_{\text{src}}, \mathbf{c}_{\text{tgt}})$, GenTS performs text-guided editing using the same diffusion backbone as in forecasting and generation, and samples an edited time series

$$\hat{\mathbf{x}} \sim p_\theta(\mathbf{x} \mid \mathbf{x}_{\text{src}}, \mathbf{c}_{\text{tgt}}),$$

where the conditional distribution $p_\theta$ is learned such that $\hat{\mathbf{x}}$ is semantically aligned with the guidance $\mathbf{c}_{\text{tgt}}$, and $\hat{\mathbf{x}}$ remains a realistic time series that preserves the parts of $\mathbf{x}_{\text{src}}$ not contradicted by $\mathbf{c}_{\text{tgt}}$.

Throughout the paper, all automatic editing metrics are computed on the triplet $(\hat{\mathbf{x}}, \mathbf{x}_{\text{src}}, \mathbf{c}_{\text{tgt}})$, and the term original time series always refers to $\mathbf{x}_{\text{src}}$ sampled from $\mathcal{D}_{\text{test}}$.

## D  MODEL ARCHITECTURE

As introduced in Sec. 3.2, our unified generative model is built upon a trainable noise estimator $\epsilon_\theta$ from a conditional diffusion model backbone. This model is capable of handling multiple tasks such as forecasting, generation, and editing within a unified architecture, and produces the estimated noise corresponding to each task. In this section, we provide a detailed description of the noise estimator architecture. See Fig. 6 for the model architecture.

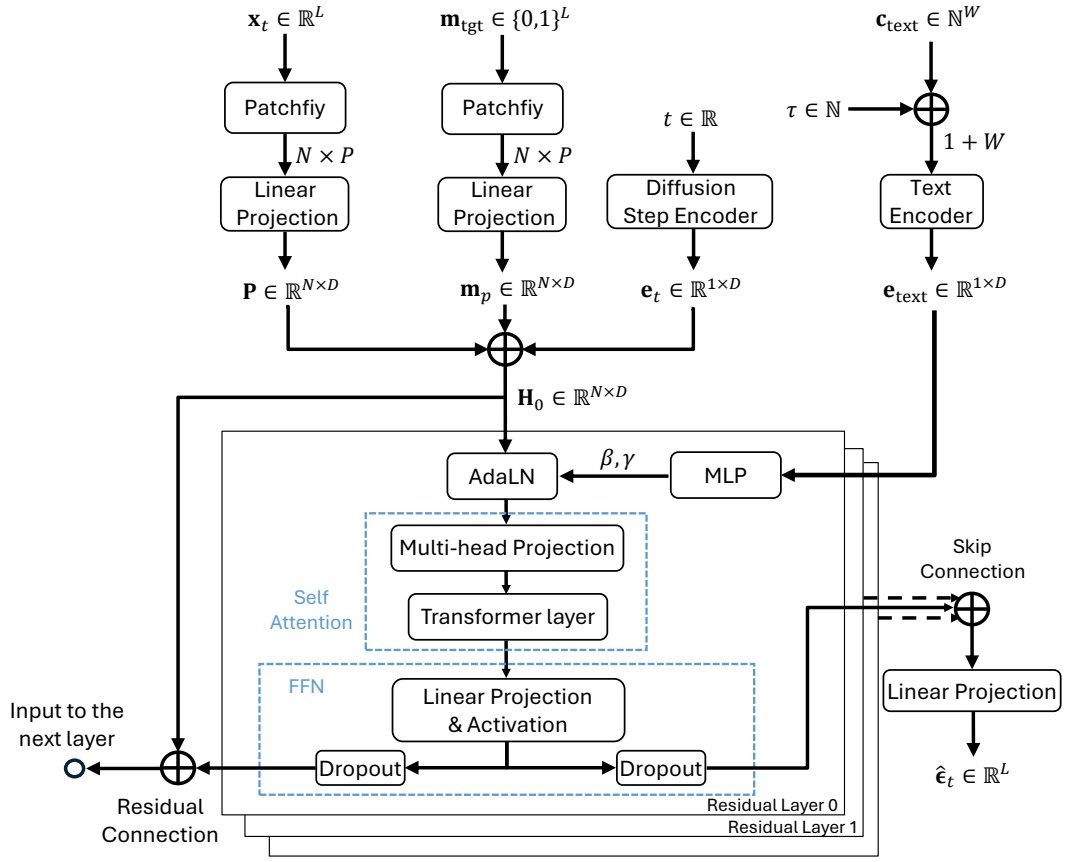

Figure 6: The model architecture of the noise estimator in GENTS.

Initially, our noise estimator $\epsilon_\theta(\mathbf{x}_t, \mathbf{c}_{\text{text}}, \tau, t)$ takes 4 inputs: the noisy time series $\mathbf{x}_t \in \mathbb{R}^L$, a text condition $\mathbf{c}_{\text{text}} \in \mathbb{N}^W$, a task type token $\tau \in \mathbb{N}$, and the diffusion step $t \in \mathbb{R}$. $L$ is the original time series length, and $W$ is the number of tokens in the text condition.

First, the original noisy time series $\mathbf{x}_t \in \mathbb{R}^L$ and the target binary mask $\mathbf{m}_{\text{tgt}} \in \{0,1\}^L$ are fed into their respective patchify module and linear encoder module, resulting in the time series embedding $\mathbf{P} \in \mathbb{R}^{N \times D}$ and the mask embedding $\mathbf{m}_p \in \mathbb{R}^{N \times D}$, respectively. At the same time, the diffusion step $t \in \mathbb{R}$ is processed by a diffusion step encoder, ultimately producing the diffusion step embedding $\mathbf{e}_t \in \mathbb{R}^{N \times D}$. On the other side, the task type token $\tau \in \mathbb{N}$ is prepended to the conditional textual description $\mathbf{c}_{\text{text}} \in \mathbb{N}^W$. The text description with the task token is then encoded to obtain the text condition embedding $\mathbf{e}_c \in \mathbb{R}^D$.

Next, these four embeddings are fed into the residual layers. Specifically, the three embeddings $\mathbf{P} \in \mathbb{R}^{N \times D}$, $\mathbf{m}_p \in \mathbb{R}^{N \times D}$, and $\mathbf{e}_t \in \mathbb{R}^{N \times D}$ are combined to form the input of the 0-th layer, $\mathbf{H}^0 = (\mathbf{P} + \mathbf{m}_p + \mathbf{e}_t) \in \mathbb{R}^{N \times D}$. The text condition embedding $\mathbf{e}_c \in \mathbb{R}^D$ is passed through an MLP layer to produce the parameters $\beta, \gamma = \text{MLP}(\mathbf{e}_c)$.

In each specific $j$-th layer of the noise estimator, the input from the previous layer $\mathbf{H}^{j-1} \in \mathbb{R}^{N \times D}$ is combined with the MLP outputs through the Adaptive Layer Normalization mechanism to yield $\tilde{\mathbf{H}}^j = (\mathbf{H}^{j-1} \times \beta + \gamma) \in \mathbb{R}^{N \times D}$, fusing time series and semantic information. $\tilde{\mathbf{H}}^j$ is then passed

to the Self-Attention (SA) module to learn the interactions among the noisy time series and the condition part of the time series. The SA module is formulated as follows:

$$\text{SA}(\tilde{\mathbf{H}}^j) = \text{softmax}\left(\frac{\mathbf{Q}\mathbf{K}^\top}{\sqrt{D}}\right)\mathbf{V},$$

where the query, key, and value matrices are computed as

$$\mathbf{Q} = \tilde{\mathbf{H}}^j \mathbf{W}_Q, \quad \mathbf{K} = \tilde{\mathbf{H}}^j \mathbf{W}_K, \quad \mathbf{V} = \tilde{\mathbf{H}}^j \mathbf{W}_V,$$

with $\mathbf{W}_Q, \mathbf{W}_K, \mathbf{W}_V \in \mathbb{R}^{D \times D}$. The output of the SA module, $\text{SA}(\tilde{\mathbf{H}}^j) \in \mathbb{R}^{N \times D}$, is then passed through a feed-forward network (FFN), and the result is added to the input of the current layer $\mathbf{H}^{j-1}$ to produce the output for the next layer:

$$\mathbf{H}^j = \text{FFN}(\text{SA}(\tilde{\mathbf{H}}^j)) + \mathbf{H}^{j-1}, \quad \mathbf{H}^j \in \mathbb{R}^{N \times D}.$$

Finally, the output of the $J$-th layer, $\mathbf{H}^J \in \mathbb{R}^{N \times D}$, is fed into a patch decoder to transfer the estimated noise into the original time series space, producing the final output $\hat{\boldsymbol{\epsilon}}_t \in \mathbb{R}^L$.

## E  CTTP MODEL

As described in Sec. 4.1, we adopt the CTTP (Contrastive Text-Time Series Pretraining) model to measure the semantic similarity between the final generated time series and the original textual conditions. In this section, we provide a detailed explanation of the CTTP model.

The CTTP model is designed to learn a shared embedding space for time series and textual descriptions, enabling a quantitative evaluation of semantic alignment between the two modalities.

Consider a batch of $B$ paired input samples $\{(\mathbf{x}_i, \mathbf{c}_i)\}_{i=1}^B$, where $\mathbf{x}_i \in \mathbb{R}^L$ denotes the $i$-th multivariate generated time series of length $L$, and $\mathbf{c}_i \in \mathbb{N}^M$ denotes the $i$-th corresponding text condition of $M$ tokens.

The CTTP model consists of two encoders: a time series encoder and a text encoder. The time series encoder, denoted as $\psi_{\text{ts}}(\cdot)$, is implemented using PatchTST (Nie et al., 2022), which is well-suited for extracting high-level representations from time series data. The text encoder, denoted as $\psi_{\text{text}}(\cdot)$, utilizes a pre-trained Long-clip (Zhang et al., 2024) model to encode the text descriptions. Both encoders project their respective inputs into a $d$-dimensional shared embedding space.

Given a batch of $B$ pairs, we obtain the embeddings for all samples:

$$\mathbf{Z}_{\text{ts}} = [\psi_{\text{ts}}(\mathbf{x}_1); \ldots; \psi_{\text{ts}}(\mathbf{x}_B)] \in \mathbb{R}^{B \times d}, \qquad \mathbf{Z}_{\text{text}} = [\psi_{\text{text}}(\mathbf{c}_1); \ldots; \psi_{\text{text}}(\mathbf{c}_B)] \in \mathbb{R}^{B \times d},$$

where $d$ is the embedding dimension, and $[\cdot; \cdot]$ denotes stacking.

To quantify the cross-modal similarity, we compute a similarity matrix $\mathbf{S} \in \mathbb{R}^{B \times B}$, where each entry is given by the (optionally normalized) dot product between the time series and text embeddings:

$$\mathbf{S}_{i,j} = \langle \mathbf{Z}_{\text{ts}}[i], \mathbf{Z}_{\text{text}}[j] \rangle,$$

where $\langle \cdot, \cdot \rangle$ denotes the inner product and $\mathbf{Z}_{\text{ts}}[i] \in \mathbb{R}^d$ is the $i$-th time series embedding, $\mathbf{Z}_{\text{text}}[j] \in \mathbb{R}^d$ is the $j$-th text embedding.

The model is trained with a symmetric contrastive objective, encouraging the embeddings of matched pairs to be closer than those of mismatched pairs. Specifically, we use a bidirectional cross-entropy loss:

$$\mathcal{L}_{\text{CTTP}} = \frac{1}{2}\left(\mathcal{L}_{\text{ts}\to\text{text}} + \mathcal{L}_{\text{text}\to\text{ts}}\right),$$

where

$$\mathcal{L}_{\text{ts}\to\text{text}} = -\frac{1}{B}\sum_{i=1}^B \log \frac{\exp(\mathbf{S}_{i,i})}{\sum_{j=1}^B \exp(\mathbf{S}_{i,j})}, \qquad \mathcal{L}_{\text{text}\to\text{ts}} = -\frac{1}{B}\sum_{j=1}^B \log \frac{\exp(\mathbf{S}_{j,j})}{\sum_{i=1}^B \exp(\mathbf{S}_{i,j})}.$$

Here, $\mathcal{L}_{\text{ts}\to\text{text}}$ enforces that the embedding of each time series is most similar to its paired text, and $\mathcal{L}_{\text{text}\to\text{ts}}$ enforces the reverse. This symmetric structure ensures robust alignment between the two modalities.

Finally, the CTTP model outputs $\mathcal{L}_{\text{CTTP}}$. Details about utilizing CTTP model to calculate CTTP Score for evaluation would be mentioned in Appendix F.

# F EVALUATION METRICS

As described in Sec. 4.1, for the forecasting task, we use MSE and MAE to assess the accuracy. For the generation task, J-FTSD and CTTP score are adopted to evaluate the fidelity and alignment, respectively. For the editing task, RaTS and CTTP are adopted to evaluate the editing performance. In the following part of this section, we introduce the details of J-FTSD, CTTP score and RaTS metrics.

## F.1 J-FTSD

In the generation task, the Joint Fréchet Time Series Distance (J-FTSD) (Narasimhan et al., 2024) is used to quantify the similarity of the joint distributions of (time series, textual condition) pairs between real time series and generated time series, thus evaluating the fidelity of conditional generation.

For a dataset of $K$ samples, let $\{(\mathbf{x}^{(i)}, \mathbf{c}_{\text{text}}^{(i)})\}_{i=1}^{K}$ be the real pairs, and $\{(\hat{\mathbf{x}}^{(i)}, \mathbf{c}_{\text{text}}^{(i)})\}_{i=1}^{K}$ the generated pairs. We extract joint embeddings by concatenating the time series and text representations:

$$\mathbf{z}^{(i)} = [\,\psi_{\text{ts}}(\mathbf{x}^{(i)}); \, \psi_{\text{text}}(\mathbf{c}_{\text{text}}^{(i)})\,], \quad \hat{\mathbf{z}}^{(i)} = [\,\psi_{\text{ts}}(\hat{\mathbf{x}}^{(i)}); \, \psi_{\text{text}}(\mathbf{c}_{\text{text}}^{(i)})\,]$$

where $\psi_{\text{ts}}(\cdot)$ and $\psi_{\text{text}}(\cdot)$ are the encoders for time series and text, and $[\cdot; \cdot]$ denotes vector concatenation.

J-FTSD is then defined as the Fréchet distance between the empirical distributions of $\{\mathbf{z}^{(i)}\}$ and $\{\hat{\mathbf{z}}^{(i)}\}$:

$$\text{J-FTSD} = \|\boldsymbol{\mu}_z - \boldsymbol{\mu}_{\hat{z}}\|_2^2 + \text{Tr}\left(\boldsymbol{\Sigma}_z + \boldsymbol{\Sigma}_{\hat{z}} - 2\left(\boldsymbol{\Sigma}_z \boldsymbol{\Sigma}_{\hat{z}}\right)^{1/2}\right)$$

where $\boldsymbol{\mu}_z$, $\boldsymbol{\Sigma}_z$ are the mean and covariance of the real joint embeddings, and $\boldsymbol{\mu}_{\hat{z}}$, $\boldsymbol{\Sigma}_{\hat{z}}$ are those of the generated joint embeddings. Smaller J-FTSD values indicate that the generated data distribution is closer to the real data in the joint embedding space.

## F.2 CTTP SCORE

In the generation task, the CTTP score measures the semantic alignment between a generated time series and the ground truth time series, based on the embedding space learned by the CTTP model. For more details about CTTP model, refer to Appendix E.

Given a generated time series $\hat{\mathbf{x}}$ and ground truth time series $\mathbf{x}$, their embeddings are obtained as $\mathbf{z}_{\text{gen}} = \psi_{\text{ts}}(\hat{\mathbf{x}})$ and $\mathbf{z}_{\text{gt}} = \psi_{\text{ts}}(\mathbf{x})$. The CTTP score is then computed as the dot product between these two embeddings:

$$\text{CTTP\_score}(\hat{\mathbf{x}}, \mathbf{x}) = \mathbf{z}_{\text{gen}}^{\top} \mathbf{z}_{\text{gt}}$$

where $\psi_{\text{ts}}(\cdot)$ is the time series encoder of the CTTP model. Higher CTTP scores indicate stronger alignment between the generated time series and the ground truth time series.

## F.3 RATS

In the editing task, Log Ratio of Target-to-Source probability (RaTS) (Jing et al., 2024a) measures whether the generated time series $\hat{\mathbf{x}}$ is closer to the given textual description $\mathbf{c}_{\text{text}}$ than the source time series $\mathbf{x}$. Formally, the RaTS for a tuple $(\hat{\mathbf{x}}, \mathbf{x}, \mathbf{c}_{\text{text}})$ is defined as:

$$\text{RaTS}(\hat{\mathbf{x}}, \mathbf{x}, \mathbf{c}_{\text{text}}) = \log\left(\frac{p(\mathbf{c}_{\text{text}}|\hat{\mathbf{x}})}{p(\mathbf{c}_{\text{text}}|\mathbf{x})}\right),$$

where $p(\mathbf{c}_{\text{text}}|\mathbf{x})$ is calculated by applying a sigmoid over the similarity score $\psi_{\text{ts}}(\mathbf{x})^{\top}\psi_{\text{text}}(\mathbf{c}_{\text{text}})$, and $\psi_{\text{ts}}, \psi_{\text{text}}$ are the time series encoder and text encoder of the CTTP model, respectively. Higher RaTS scores indicate that the edited time series $\hat{\mathbf{x}}$ is closer to the textual description $\mathbf{c}_{\text{text}}$.

## G MORE EXPERIMENTAL RESULTS

### G.1 QUANTITATIVE RESULTS

We report forecasting results across multiple horizons on six datasets, as shown in Tab. 7. Additionally, we present generation and editing results on the same six datasets in Tab. 8. All experiments are conducted three times with different random seeds, and we report the mean and standard deviation of the results. These results prove the effectiveness of GENTS across forecasting, generation, and editing.

Table 7: Forecasting performance of different horizons on all datasets with standard deviation (mean±std). The best performance is with bold font, and the second-best results are underlined.

| Dataset | Horizon | Metric | Dlinear | Autoformer | TimesNet | Diffusion-TS | Chronos | Moirai | TimeCMA | TimeMMD | IATSF | GENTS (ours) |
|---|---|---|---|---|---|---|---|---|---|---|---|---|
| TimeMMD-Env | 48 | MAE | $0.546_{\pm0.000}$ | $0.988_{\pm0.149}$ | $\underline{0.498}_{\pm0.003}$ | $0.527_{\pm0.001}$ | $0.566_{\pm0.001}$ | $0.620_{\pm0.002}$ | $0.507_{\pm0.001}$ | $\underline{0.481}_{\pm0.006}$ | $\mathbf{0.478}_{\pm0.001}$ | $\underline{0.481}_{\pm0.004}$ |
| | | MSE | $0.494_{\pm0.000}$ | $1.595_{\pm0.744}$ | $0.484_{\pm0.007}$ | $0.568_{\pm0.002}$ | $0.642_{\pm0.003}$ | $1.057_{\pm0.022}$ | $0.492_{\pm0.000}$ | $\mathbf{0.426}_{\pm0.004}$ | $0.456_{\pm0.001}$ | $\underline{0.469}_{\pm0.017}$ |
| | 96 | MAE | $0.641_{\pm0.000}$ | $0.721_{\pm0.014}$ | $0.550_{\pm0.010}$ | $0.534_{\pm0.002}$ | $0.654_{\pm0.002}$ | $0.755_{\pm0.004}$ | $0.547_{\pm0.001}$ | $\mathbf{0.513}_{\pm0.006}$ | $0.523_{\pm0.005}$ | $\underline{0.522}_{\pm0.037}$ |
| | | MSE | $0.610_{\pm0.000}$ | $0.889_{\pm0.050}$ | $0.578_{\pm0.019}$ | $0.590_{\pm0.002}$ | $0.832_{\pm0.008}$ | $1.693_{\pm0.128}$ | $0.558_{\pm0.001}$ | $\mathbf{0.480}_{\pm0.009}$ | $0.535_{\pm0.003}$ | $\underline{0.533}_{\pm0.053}$ |
| | 144 | MAE | $0.694_{\pm0.000}$ | $0.796_{\pm0.021}$ | $0.564_{\pm0.010}$ | $0.537_{\pm0.004}$ | $0.683_{\pm0.002}$ | $0.832_{\pm0.004}$ | $0.548_{\pm0.000}$ | $\mathbf{0.523}_{\pm0.010}$ | $0.533_{\pm0.004}$ | $\underline{0.524}_{\pm0.043}$ |
| | | MSE | $0.682_{\pm0.000}$ | $1.170_{\pm0.125}$ | $0.605_{\pm0.017}$ | $0.603_{\pm0.005}$ | $0.896_{\pm0.010}$ | $2.138_{\pm0.096}$ | $0.565_{\pm0.000}$ | $\mathbf{0.496}_{\pm0.006}$ | $0.556_{\pm0.006}$ | $\underline{0.522}_{\pm0.054}$ |
| Weather | 36 | MAE | $0.226_{\pm0.002}$ | $0.395_{\pm0.028}$ | $0.197_{\pm0.004}$ | $0.314_{\pm0.003}$ | $0.196_{\pm0.001}$ | $0.261_{\pm0.000}$ | $0.299_{\pm0.000}$ | $0.265_{\pm0.003}$ | $\underline{0.183}_{\pm0.002}$ | $\mathbf{0.147}_{\pm0.002}$ |
| | | MSE | $0.124_{\pm0.001}$ | $0.286_{\pm0.032}$ | $0.114_{\pm0.003}$ | $0.402_{\pm0.003}$ | $0.112_{\pm0.001}$ | $0.168_{\pm0.000}$ | $0.190_{\pm0.000}$ | $0.130_{\pm0.002}$ | $\underline{0.092}_{\pm0.001}$ | $\mathbf{0.076}_{\pm0.001}$ |
| | 72 | MAE | $0.319_{\pm0.001}$ | $0.450_{\pm0.062}$ | $0.310_{\pm0.001}$ | $0.460_{\pm0.002}$ | $0.308_{\pm0.001}$ | $0.364_{\pm0.000}$ | $0.372_{\pm0.000}$ | $0.334_{\pm0.002}$ | $\underline{0.276}_{\pm0.001}$ | $\mathbf{0.226}_{\pm0.000}$ |
| | | MSE | $0.220_{\pm0.001}$ | $0.388_{\pm0.093}$ | $0.221_{\pm0.001}$ | $0.391_{\pm0.003}$ | $0.234_{\pm0.000}$ | $0.298_{\pm0.000}$ | $0.285_{\pm0.000}$ | $0.214_{\pm0.002}$ | $\underline{0.172}_{\pm0.002}$ | $\mathbf{0.139}_{\pm0.001}$ |
| | 108 | MAE | $0.368_{\pm0.001}$ | $0.487_{\pm0.016}$ | $0.365_{\pm0.000}$ | $0.487_{\pm0.001}$ | $0.372_{\pm0.000}$ | $0.421_{\pm0.000}$ | $0.413_{\pm0.000}$ | $0.379_{\pm0.000}$ | $\underline{0.342}_{\pm0.000}$ | $\mathbf{0.289}_{\pm0.001}$ |
| | | MSE | $0.271_{\pm0.001}$ | $0.456_{\pm0.045}$ | $0.286_{\pm0.004}$ | $0.425_{\pm0.001}$ | $0.305_{\pm0.001}$ | $0.375_{\pm0.000}$ | $0.335_{\pm0.000}$ | $\underline{0.265}_{\pm0.002}$ | $0.243_{0.000}$ | $\mathbf{0.201}_{\pm0.002}$ |
| HongKong | 28 | MAE | $0.891_{\pm0.014}$ | $1.468_{\pm0.313}$ | $0.710_{\pm0.003}$ | $0.760_{\pm0.006}$ | $0.727_{\pm0.002}$ | $0.799_{\pm0.002}$ | $0.705_{\pm0.003}$ | $\mathbf{0.697}_{\pm0.002}$ | $\underline{0.704}_{\pm0.002}$ | $0.725_{\pm0.013}$ |
| | | MSE | $1.376_{\pm0.039}$ | $3.246_{\pm1.207}$ | $0.958_{\pm0.013}$ | $1.099_{\pm0.016}$ | $1.049_{\pm0.003}$ | $1.254_{\pm0.006}$ | $0.927_{\pm0.007}$ | $\mathbf{0.911}_{\pm0.006}$ | $\underline{0.922}_{\pm0.006}$ | $0.978_{\pm0.038}$ |
| | 56 | MAE | $0.926_{\pm0.010}$ | $1.808_{\pm0.278}$ | $0.782_{\pm0.002}$ | $0.811_{\pm0.009}$ | $0.795_{\pm0.002}$ | $0.892_{\pm0.001}$ | $0.777_{\pm0.001}$ | $0.748_{\pm0.002}$ | $\mathbf{0.714}_{\pm0.030}$ | $\underline{0.744}_{\pm0.023}$ |
| | | MSE | $1.447_{\pm0.031}$ | $4.863_{\pm1.185}$ | $1.081_{\pm0.008}$ | $1.207_{\pm0.032}$ | $1.170_{\pm0.005}$ | $1.458_{\pm0.008}$ | $1.047_{\pm0.002}$ | $\underline{0.988}_{\pm0.004}$ | $\mathbf{0.915}_{\pm0.079}$ | $0.997_{\pm0.044}$ |
| | 84 | MAE | $0.948_{\pm0.029}$ | $2.222_{\pm0.589}$ | $0.865_{\pm0.003}$ | $0.887_{\pm0.010}$ | $0.871_{\pm0.001}$ | $0.971_{\pm0.000}$ | $0.859_{\pm0.001}$ | $0.817_{\pm0.001}$ | $\underline{0.787}_{\pm0.041}$ | $\mathbf{0.762}_{\pm0.037}$ |
| | | MSE | $1.475_{\pm0.080}$ | $8.407_{\pm2.136}$ | $1.285_{\pm0.006}$ | $1.394_{\pm0.011}$ | $1.357_{\pm0.005}$ | $1.661_{\pm0.004}$ | $1.242_{\pm0.001}$ | $1.150_{\pm0.005}$ | $\underline{1.085}_{\pm0.117}$ | $\mathbf{1.034}_{\pm0.083}$ |
| ETTm1 | 48 | MAE | $0.825_{\pm0.010}$ | $1.345_{\pm0.277}$ | $0.881_{\pm0.010}$ | $\underline{0.810}_{\pm0.005}$ | $0.827_{\pm0.001}$ | $1.249_{\pm0.225}$ | $0.875_{\pm0.001}$ | $0.818_{\pm0.023}$ | $0.838_{\pm0.038}$ | $\mathbf{0.791}_{\pm0.047}$ |
| | | MSE | $1.441_{\pm0.065}$ | $3.177_{\pm1.391}$ | $1.651_{\pm0.030}$ | $\mathbf{1.286}_{\pm0.019}$ | $1.469_{\pm0.009}$ | $1.860_{\pm0.002}$ | $1.609_{\pm0.006}$ | $1.530_{\pm0.053}$ | $1.476_{\pm0.118}$ | $\underline{1.403}_{\pm0.099}$ |
| | 96 | MAE | $\mathbf{0.637}_{\pm0.006}$ | $1.371_{\pm0.303}$ | $0.716_{\pm0.030}$ | $0.691_{\pm0.004}$ | $0.730_{\pm0.001}$ | $1.863_{\pm0.140}$ | $0.718_{\pm0.001}$ | $\underline{0.640}_{\pm0.019}$ | $0.702_{\pm0.010}$ | $0.665_{\pm0.036}$ |
| | | MSE | $\mathbf{0.959}_{\pm0.029}$ | $3.070_{\pm0.993}$ | $1.183_{\pm0.127}$ | $\underline{0.964}_{\pm0.004}$ | $1.158_{\pm0.003}$ | $1.621_{\pm0.004}$ | $1.114_{\pm0.002}$ | $0.994_{\pm0.041}$ | $1.046_{\pm0.041}$ | $0.987_{\pm0.089}$ |
| | 144 | MAE | $\mathbf{0.704}_{\pm0.001}$ | $1.054_{\pm0.171}$ | $0.773_{\pm0.026}$ | $0.736_{\pm0.008}$ | $0.758_{\pm0.002}$ | $1.152_{\pm0.015}$ | $0.779_{\pm0.000}$ | $0.715_{\pm0.015}$ | $0.762_{\pm0.007}$ | $\underline{0.711}_{\pm0.047}$ |
| | | MSE | $1.124_{\pm0.005}$ | $2.197_{\pm0.557}$ | $1.307_{\pm0.083}$ | $\mathbf{1.070}_{\pm0.020}$ | $1.203_{\pm0.006}$ | $1.773_{\pm0.003}$ | $1.293_{\pm0.004}$ | $1.181_{\pm0.033}$ | $1.213_{\pm0.021}$ | $\underline{1.115}_{\pm0.090}$ |
| Exchange | 48 | MAE | $0.225_{\pm0.018}$ | $1.023_{\pm0.485}$ | $0.147_{\pm0.000}$ | $0.301_{\pm0.006}$ | $0.135_{\pm0.002}$ | $0.132_{\pm0.000}$ | $0.152_{\pm0.000}$ | $0.134_{\pm0.002}$ | $\mathbf{0.127}_{\pm0.001}$ | $\underline{0.131}_{\pm0.005}$ |
| | | MSE | $0.110_{\pm0.018}$ | $1.771_{\pm1.489}$ | $0.040_{\pm0.000}$ | $0.186_{\pm0.005}$ | $0.038_{\pm0.001}$ | $\underline{0.037}_{\pm0.002}$ | $0.043_{\pm0.000}$ | $0.043_{\pm0.001}$ | $0.043_{\pm0.001}$ | $\mathbf{0.033}_{\pm0.003}$ |
| | 96 | MAE | $0.255_{\pm0.008}$ | $0.637_{\pm0.166}$ | $0.185_{\pm0.004}$ | $0.324_{\pm0.006}$ | $\underline{0.179}_{\pm0.003}$ | $0.183_{\pm0.000}$ | $0.185_{\pm0.000}$ | $0.177_{\pm0.002}$ | $\mathbf{0.169}_{\pm0.001}$ | $0.186_{\pm0.005}$ |
| | | MSE | $0.128_{\pm0.009}$ | $0.805_{\pm0.208}$ | $\underline{0.064}_{\pm0.003}$ | $0.198_{\pm0.003}$ | $\underline{0.064}_{\pm0.003}$ | $0.071_{\pm0.003}$ | $\mathbf{0.063}_{\pm0.000}$ | $0.065_{\pm0.002}$ | $0.067_{\pm0.001}$ | $0.066_{\pm0.003}$ |
| | 144 | MAE | $0.279_{\pm0.000}$ | $1.187_{\pm0.708}$ | $0.217_{\pm0.003}$ | $0.343_{\pm0.003}$ | $\underline{0.215}_{\pm0.003}$ | $0.226_{\pm0.000}$ | $0.218_{\pm0.000}$ | $0.212_{\pm0.001}$ | $\mathbf{0.202}_{\pm0.001}$ | $0.229_{\pm0.004}$ |
| | | MSE | $0.148_{\pm0.003}$ | $2.809_{\pm2.217}$ | $\mathbf{0.087}_{\pm0.002}$ | $0.211_{\pm0.002}$ | $0.092_{\pm0.004}$ | $0.100_{\pm0.022}$ | $\underline{0.088}_{\pm0.000}$ | $0.089_{\pm0.001}$ | $0.088_{\pm0.002}$ | $0.097_{\pm0.004}$ |
| Traffic | 48 | MAE | $0.538_{\pm0.004}$ | $3.268_{\pm0.402}$ | $0.404_{\pm0.016}$ | $0.514_{\pm0.001}$ | $0.615_{\pm0.001}$ | $1.126_{\pm0.003}$ | $0.926_{\pm0.000}$ | $\underline{0.389}_{\pm0.004}$ | $0.906_{\pm0.001}$ | $\mathbf{0.344}_{\pm0.005}$ |
| | | MSE | $0.456_{\pm0.005}$ | $13.996_{\pm4.528}$ | $0.281_{\pm0.016}$ | $0.490_{\pm0.002}$ | $0.674_{\pm0.003}$ | $1.973_{\pm0.011}$ | $1.071_{\pm0.000}$ | $\underline{0.262}_{\pm0.003}$ | $1.024_{\pm0.001}$ | $\mathbf{0.258}_{\pm0.005}$ |
| | 96 | MAE | $0.582_{\pm0.002}$ | $3.445_{\pm0.431}$ | $\underline{0.504}_{\pm0.010}$ | $0.580_{\pm0.006}$ | $0.678_{\pm0.001}$ | $1.164_{\pm0.000}$ | $0.927_{\pm0.000}$ | $\mathbf{0.459}_{\pm0.017}$ | $0.917_{\pm0.001}$ | $\underline{0.488}_{\pm0.021}$ |
| | | MSE | $0.519_{\pm0.003}$ | $15.312_{\pm2.764}$ | $\mathbf{0.396}_{\pm0.010}$ | $0.619_{\pm0.010}$ | $0.757_{\pm0.002}$ | $2.137_{\pm0.002}$ | $1.076_{\pm0.000}$ | $0.454_{\pm0.017}$ | $1.036_{\pm0.004}$ | $\underline{0.465}_{\pm0.032}$ |
| | 144 | MAE | $0.606_{\pm0.003}$ | $2.114_{\pm0.996}$ | $0.551_{\pm0.003}$ | $0.632_{\pm0.006}$ | $0.676_{\pm0.001}$ | $1.180_{\pm0.001}$ | $0.926_{\pm0.000}$ | $\underline{0.548}_{\pm0.015}$ | $0.926_{\pm0.004}$ | $\mathbf{0.537}_{\pm0.022}$ |
| | | MSE | $0.557_{\pm0.004}$ | $7.742_{\pm5.524}$ | $0.466_{\pm0.009}$ | $0.720_{\pm0.010}$ | $0.753_{\pm0.002}$ | $2.217_{\pm0.002}$ | $1.070_{\pm0.000}$ | $0.553_{\pm0.024}$ | $1.045_{\pm0.007}$ | $\underline{0.537}_{\pm0.036}$ |

### G.2 CASE STUDY

In this section, we present several visualization results of GENTS across six datasets and three tasks. The results are presented in Fig. 7. These results demonstrate that GENTS is capable of producing reasonable time series across diverse datasets and tasks.

To provide a deeper understanding of the editing capability, we present specific editing examples in Fig. 8. For each case, we visualize the original time series, the textual instruction, and the edited result. This confirms that GENTS effectively understands the semantic attributes in the caption and applies precise modifications to the time series structure.

### G.3 EFFICIENCY ANALYSIS

In this section, we present the efficiency comparison of GENTS with other baselines. Specifically, we compared model size and average inference time per sample on the Weather dataset using a single NVIDIA A40 GPU (batch size = 1). As shown in the Tab. 9a, although our model incurs a higher inference cost than non-diffusion models, it outperforms both foundation and diffusion baselines. Given its significant forecasting improvements, the increase in model size and the reduction in inference speed are considered acceptable trade-offs. We further analyze the GPU memory requirements

Table 8: Generation and editing performance on all datasets with standard deviation (mean±std). The best performance is with bold font. Arrows ↑(↓) indicate that higher (lower) values are better.

| Generation | TimeMMD-Env | | Weather | | HongKong | |
|---|---|---|---|---|---|---|
| | ↓JFTSD | ↑CTTP | ↓JFTSD | ↑CTTP | ↓JFTSD | ↑CTTP |
| TimeWeaver (Narasimhan et al., 2024) | $421.145_{\pm0.548}$ | $79.728_{\pm0.656}$ | $291.369_{\pm1.069}$ | $55.809_{\pm0.480}$ | $458.184_{\pm1.689}$ | $120.509_{\pm2.762}$ |
| TEdit (Jing et al., 2024a) | $423.662_{\pm0.631}$ | $76.808_{\pm1.583}$ | $290.678_{\pm0.263}$ | $54.996_{\pm0.172}$ | $457.276_{\pm2.587}$ | $120.956_{\pm2.702}$ |
| GENTS (ours) | $\mathbf{166.379}_{\pm9.230}$ | $\mathbf{83.687}_{\pm1.843}$ | $\mathbf{1.320}_{\pm0.034}$ | $\mathbf{115.251}_{\pm0.421}$ | $\mathbf{23.516}_{\pm0.317}$ | $\mathbf{140.052}_{\pm1.470}$ |

| Generation | ETTm1 | | Exchange | | Traffic | |
|---|---|---|---|---|---|---|
| | ↓JFTSD | ↑CTTP | ↓JFTSD | ↑CTTP | ↓JFTSD | ↑CTTP |
| TimeWeaver (Narasimhan et al., 2024) | $40.281_{\pm9.823}$ | $93.554_{\pm9.008}$ | $78.282_{\pm4.493}$ | $85.068_{\pm1.280}$ | $\mathbf{140.031}_{\pm4.950}$ | $\mathbf{55.520}_{\pm1.369}$ |
| TEdit (Jing et al., 2024a) | $30.761_{\pm4.660}$ | $108.750_{\pm2.602}$ | $80.606_{\pm2.259}$ | $82.621_{\pm4.064}$ | $141.305_{\pm1.824}$ | $55.097_{\pm0.526}$ |
| GENTS (ours) | $\mathbf{27.227}_{\pm3.240}$ | $\mathbf{127.575}_{\pm2.867}$ | $\mathbf{45.801}_{\pm3.021}$ | $\mathbf{180.377}_{\pm1.499}$ | $178.937_{\pm2.641}$ | $44.942_{\pm0.175}$ |

| Editing | TimeMMD-Env | | Weather | | HongKong | |
|---|---|---|---|---|---|---|
| | ↑RaTS | ↑CTTP | ↑RaTS | ↑CTTP | ↑RaTS | ↑CTTP |
| TimeWeaver (Narasimhan et al., 2024) | $0.584_{\pm0.013}$ | $79.238_{\pm0.817}$ | $0.364_{\pm0.015}$ | $56.035_{\pm0.958}$ | $0.088_{\pm0.015}$ | $117.900_{\pm1.023}$ |
| TEdit (Jing et al., 2024a) | $0.556_{\pm0.019}$ | $76.011_{\pm1.267}$ | $0.370_{\pm0.012}$ | $55.962_{\pm0.528}$ | $0.078_{\pm0.025}$ | $117.916_{\pm1.012}$ |
| GENTS (ours) | $\mathbf{1.451}_{\pm0.076}$ | $\mathbf{84.059}_{\pm2.291}$ | $\mathbf{3.071}_{\pm0.066}$ | $\mathbf{115.281}_{\pm0.304}$ | $\mathbf{0.121}_{\pm0.031}$ | $\mathbf{140.376}_{\pm1.106}$ |

| Editing | ETTm1 | | Exchange | | Traffic | |
|---|---|---|---|---|---|---|
| | ↑RaTS | ↑CTTP | ↑RaTS | ↑CTTP | ↑RaTS | ↑CTTP |
| TimeWeaver (Narasimhan et al., 2024) | $0.282_{\pm0.120}$ | $93.854_{\pm7.822}$ | $-1.411_{\pm0.250}$ | $88.967_{\pm1.527}$ | $-0.622_{\pm0.240}$ | $52.182_{\pm8.273}$ |
| TEdit (Jing et al., 2024a) | $0.551_{\pm0.033}$ | $107.843_{\pm1.443}$ | $-1.577_{\pm0.141}$ | $83.941_{\pm6.973}$ | $-0.498_{\pm0.055}$ | $\mathbf{56.721}_{\pm1.209}$ |
| GENTS (ours) | $\mathbf{0.795}_{\pm0.062}$ | $\mathbf{128.247}_{\pm3.180}$ | $\mathbf{0.796}_{\pm0.043}$ | $\mathbf{180.166}_{\pm0.609}$ | $\mathbf{4.412}_{\pm0.143}$ | $43.434_{\pm1.800}$ |

Table 9: Model efficiency and memory usage on Weather dataset.

(a) Model size and inference time.

| Method | GENTS | DiffusionTS | Chronos | TimeMMD |
|---|---|---|---|---|
| Model size (MB) | 26 | 5.8 | 769 | 35 |
| Inference time (ms) | 347 | 1571 | 691 | 17 |

(b) Peak training-time GPU memory usage (MB).

| Model | Prediction horizon | | |
|---|---|---|---|
| | 36 | 72 | 108 |
| GENTS | 7273 | 7593 | 7860 |

on the Weather dataset. With batch size = 1024, increasing the prediction horizon from 36 to 108 time steps only raises the peak training-time GPU memory from 7273 MB to 7860 MB (Tab. 9b), indicating moderate training requirements and smooth scalability with respect to sequence length.

### G.4 HUMAN EVALUATION ON TIME SERIES GENERATION

To complement the automatic metrics and further verify the effectiveness of GENTS, we conducted a human preference study focusing on two key aspects: Fidelity and Instruction Alignment.

**Settings.** We randomly sampled 20 distinct textual descriptions from the test sets. For each description, we generated time series using three models: TimeWeaver (Narasimhan et al., 2024), TEdit (Jing et al., 2024a), and our GENTS. Three human evaluators participated in this study. For every description, each evaluator was presented with the three anonymized generated samples in a randomized order, and was asked to independently rank the three samples (1st, 2nd, 3rd) based on their Fidelity and Instruction Alignment, respectively. In total, this yields 60 rank scores per model and per criterion. We report the mean and standard deviation of these ranks (lower is better).

**Results.** The quantitative results are summarized in Tab. 10. For Fidelity, all methods obtain similar average ranks and relatively large standard deviations, suggesting that assessing the visual realism of generated time series is difficult for humans and leads to noticeable disagreement across evaluators. In contrast, Instruction Alignment is easier to evaluate. Our GENTS achieves a clearly lower average rank than both baselines, indicating that the proposed unified generative framework better captures the semantic intent of textual descriptions.

Table 10: Human evaluation results (Average Rank ± Std., the lower the better) comparing GENTS with baselines.

| Model | Fidelity Rank (↓) | Instr. Align. Rank (↓) |
|---|---|---|
| TimeWeaver | $2.033_{\pm0.184}$ | $2.100_{\pm0.082}$ |
| TEdit | $2.000_{\pm0.308}$ | $2.267_{\pm0.165}$ |
| GENTS (Ours) | $\mathbf{1.967}_{\pm0.409}$ | $\mathbf{1.633}_{\pm0.155}$ |

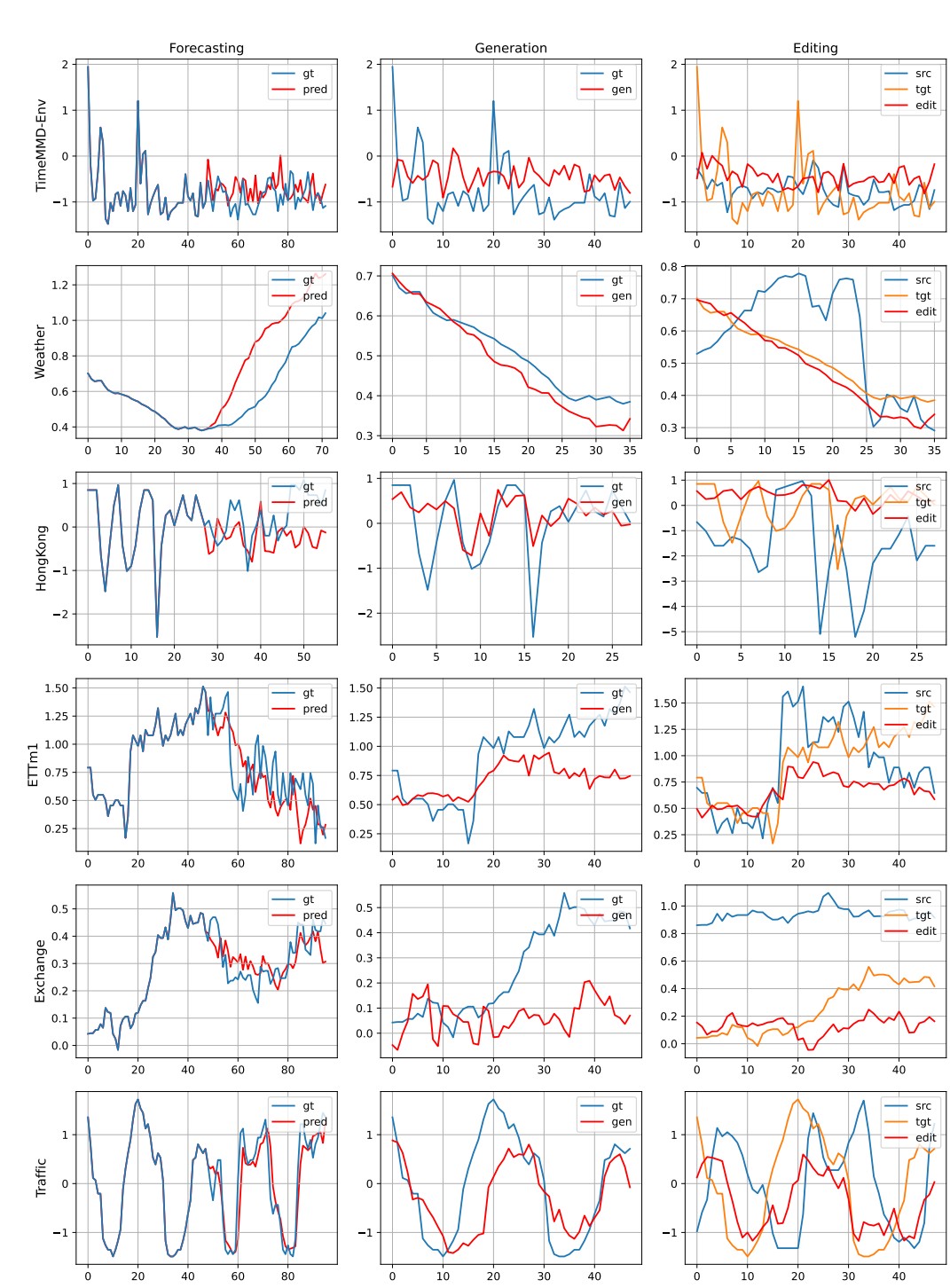

Figure 7: The case study of three tasks on six datasets. From up to down, there are TimeMMD-Env, Weather, HongKong, ETTm1, Exchange, and Traffic. From left to right, there are forecasting, generation and editing. In the legend, gt is ground truth, pred is the forecasting result, gen is the generation result, src is the original time series, tgt is the editing target, and edit is the edited result.

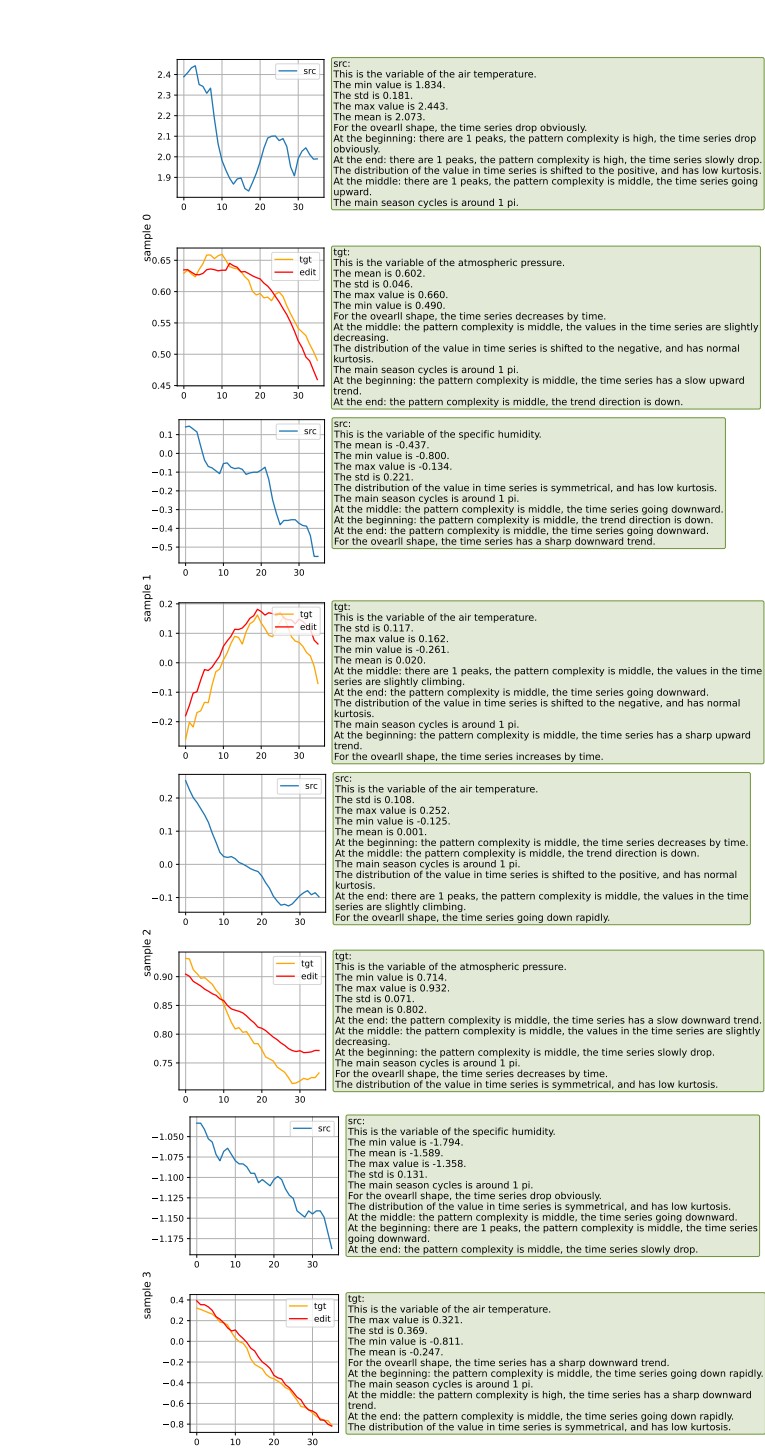

Figure 8: Additional case study for time series editing. The model takes an original time series and a textual caption as input to generate the edited result, where src is the original time series, tgt is the editing target, and edit is the editing result.

