# OpenReview forum: "Unified Generative Modeling for Multimodal Time Series Analysis"
_ICLR.cc/2026/Conference — Submitted to ICLR 2026_

### Official Review · Reviewer_woXw · 2025-10-23

**Soundness:** 3
**Presentation:** 3
**Contribution:** 3
**Rating:** 6
**Confidence:** 4

**Summary:**

Proposes GENTS, a unified generative framework for multimodal time series analysis, tackling data scarcity with domain-agnostic captioning and enabling multi-task (forecasting, generation, editing) via diffusion models.

**Strengths:**

- Introduces a flexible unified framework generalizing across multiple time series tasks (forecasting, generation, and editing).
- Domain-agnostic captioning effectively mitigates multimodal data scarcity.
- Comprehensive experiments across diverse datasets/tasks show strong performance.

**Weaknesses:**

- The ablation studies lack granularity regarding critical components. The paper needs detailed analysis of: (a) relative contributions of condition adapters versus self-attention mechanisms, (b) impact of different attribute categories in captioning, and (c) sensitivity to diffusion step embeddings.
- While Table 9 mentions inference times, the paper lacks analysis of training complexity, memory requirements, and scalability with sequence length.

**Questions:**

- How does GENTS scale computationally with increasing sequence length and variable dimensions? What is the theoretical complexity compared to task-specific baselines?
- What are the failure modes of the domain-agnostic captioning approach? Can it capture complex patterns like multi-scale anomalies or non-stationary transitions that lack clear statistical signatures?
- How does the unified training objective balance potentially conflicting optimization requirements between forecasting and generation tasks? The paper mentions λF:λG=2:1 works well, but provides no theoretical justification.

---

> ### Author Response · Authors · 2025-11-24
> **Rebuttal by Authors (part 1 of 4)**
>
> > **W1:** The ablation studies lack granularity regarding critical components. The paper needs detailed analysis of: (a) relative contributions of condition adapters versus self-attention mechanisms, (b) impact of different attribute categories in captioning, and \(c) sensitivity to diffusion step embeddings.
>
> > W1.1 (a) relative contributions of condition adapters versus self-attention mechanisms
>
> We thank the reviewer for the advice of this ablation study.
>
> We believe that these two modules are inherent components of the Transformer architecture rather than incremental designs specific to our method. Therefore, we prioritized ablations that more directly probe our main contributions.
>
> The condition adapters and self-attention mechanisms are necessary parts of our model backbone.
> - Self-attention is the core mechanism of the Transformer [1] for modeling temporal dependencies. Removing self-attention would collapse the model into a much weaker MLP-like architecture that cannot capture temporal structure.
> - Condition adapters are widely used modules for injecting conditions into diffusion backbones [2]. Without them, the model would be unable to incorporate textual information.
>
> Direct ablations of these two modules would either be degenerate (self-attention) or redundant with our existing ablations without text (condition adapter).
> - Ablating self-attention would reduce the model to a feed-forward network and eliminate its ability to model temporal dependencies, which is essential for time series tasks.
> - For condition adapters, removing them would be nearly equivalent to removing textual conditioning. In practice, we already evaluate this effect by ablating textual conditions in Table 3 of our paper. These results show that removing text leads to a clear performance drop in both forecasting and generation. Since the adapters are precisely the mechanism through which textual conditions are injected, we expect an explicit no-adapter variant to exhibit similar degradation.
>
> [1] Attention Is All You Need. NeurIPS 2017.
>
> [2] Scalable Diffusion Models with Transformers. ICCV 2023.
>
> > W1.2 (b) impact of different attribute categories in captioning
>
> We thank the reviewer for asking about the impact of different attribute categories in our captioning.
>
> We clarify that this analysis is already included in Fig.3 of our paper.
> - Experiment setting. We categorized the descriptions in the domain-agnostic captions into different attribute groups and independently masked each group. The performance drop in forecasting after masking is used to quantify the impact of each attribute category.
> - Experiment results. As shown in Fig.3, all extracted attribute categories contribute to forecasting performance improvements across multiple datasets, indicating that each attribute category in the domain-agnostic caption is meaningful and beneficial.
>
> > W1.3 \(c) sensitivity to diffusion step embeddings.
>
> We thank the reviewer for raising this question, and we agree that diffusion step embeddings play an important role in the sampling stage.
>
> Diffusion step embeddings are explicitly modeled in GenTS which follow standard diffusion practices. Our understanding of the reviewer's concern is whether the diffusion step is explicitly considered during noise estimation. This design aligns with common diffusion model practice, where timestep embeddings are essential for conditioning the denoiser on the current diffusion stage [3, 4]. Intuitively, the timestep embedding informs the network how much noise remains and how aggressively it should denoise at step $t$, which is crucial for stable multi-step sampling.
>
> Per your suggestions, we have conducted additional experiments on it and we found that *ablating diffusion step embeddings causes a significant drop in performance*. To assess the sensitivity to diffusion step embeddings, we performed an ablation in which all timestep embeddings are removed and the model is trained to predict noise without explicit step conditioning. The results are shown in Table 1 below. Across diverse datasets and tasks, we observe a consistent and substantial degradation. This confirms that diffusion step is necessary in diffusion model.
>
> [3] Denoising Diffusion Probabilistic Models. NeurIPS 2020.
>
> [4] Improved Denoising Diffusion Probabilistic Models. ICML 2021.
>
> #### Table 1. Ablation study on diffusion step embeddings. $\uparrow$ ($\downarrow$) means the higher (lower), the better.
>
> |Task|Forecasting|Forecasting|Forecasting|Forecasting|Geneartion|Geneartion|Geneartion|Geneartion|
> |:--:|:--:|:--:|:--:|:--:|:--:|:--:|:--:|:--:|
> |Dataset|HongKong|HongKong|Traffic|Traffic|HongKong|HongKong|Traffic|Traffic|
> |Metric|$\downarrow$MAE|$\downarrow$MSE|$\downarrow$MAE|$\downarrow$MSE|$\downarrow$JFTSD|$\uparrow$CTTP|$\downarrow$JFTSD|$\uparrow$CTTP|
> |GenTS|0.743|1.003|0.456|0.420|23.516|140.052|178.937|44.942|
> |w/o diffusion step embedding|2.597|20.546|1.047|1.963|62.772|115.894|201.427|33.449|

---

> ### Author Response · Authors · 2025-11-24
> **Rebuttal by Authors (part 2 of 4)**
>
> > **W2:** While Table 9 mentions inference times, the paper lacks analysis of training complexity, memory requirements, and scalability with sequence length.
>
> We thank the reviewer for this suggestion, and we add some analysis about training complexity, memory requirements, and scalability with sequence length in this response.
>
> (i) Training complexity.
> Let the input time series have length $L$ and patch size $P$, so the number of time-series patches is $N=L/P$. We denote by $W$ the number of text tokens and by $D$ the hidden dimension. Under these definitions, the training-time complexity of GenTS is dominated by a $J\_1$-layer Transformer-based noise estimator whose cost scales with the number of time-series patches $N$, and a $J\_2$-layer Transformer-based text encoder whose cost scales with the text length $W$.
> - Forward pass. Each layer of the noise estimator operates on the $N$ time-series patches, yielding the standard Transformer cost $O(N^2D + ND^2)$ per layer, while each layer of the text encoder over the $W$ text tokens incurs $O(W^2D + WD^2)$. Therefore, the overall forward complexity per training sample is $O(J\_1(N^2D + ND^2) + J\_2(W^2D + WD^2))$, where the first term comes from the noise estimator over time-series patches and the second term from the text encoder.
> - Backward pass. The backward pass has the same asymptotic complexity as the forward pass. Since gradients must be propagated through all layers of the noise estimator and the text encoder, the backward cost is only a constant-factor multiple of the forward cost.
>
> Combining forward and backward, the per-step training time is $O(J\_1(N^2D + ND^2) + J\_2(W^2D + WD^2))$, which matches the standard training complexity of Transformer-based diffusion models [5, 6].
>
> (ii) Memory requirements, and scalability with sequence length.
> We conduct an experiment on Weather dataset to analyze the GPU memory requirements of GenTS and how they vary with sequence length. We fix the batch size to 1024 and keep the model configuration and hardware unchanged, while fixing the history length to 36 time steps and varying the forecast horizon over {36, 72, 108} time steps. The peak training-time GPU memory usage is reported in Table 2 below.
> From Table 2 we observe that increasing the prediction horizon from 36 to 108 only raises the peak memory usage from 7273 MB to 7860 MB, which is an increase of about 8% under the same batch size and architecture. This suggests that, in our implementation, the dominant contributors to memory consumption during training are the text-related components which do not depend on the time-series sequence length. In contrast, the additional activations introduced by longer time-series inputs, contribute only a relatively small overhead in this experimental setting.
> Overall, within the sequence-length range considered in our benchmarks, our GenTS has manageable memory requirements and scales smoothly with respect to the prediction horizon.
>
> In summary, both the theoretical complexity and the measured memory usage of GenTS are moderate, which makes it practical to train GenTS end-to-end on a single GPU with a large batch size (1024) on real-world datasets such as Weather. We will add this memory usage experiment in the revised version.
>
> #### Table 2. Memory usage of GenTS on three prediction horizons of Weather dataset.
> |Prediction Horizon|36|72|108|
> |:--:|:--:|:--:|:--:|
> |Memory usage (MB)|7273|7593|7860|
>
> [5] Diffusion-TS: Interpretable Diffusion for General Time Series Generation. ICLR 2024.
>
> [6] Towards Editing Time Series. NeurIPS 2024.

---

> ### Author Response · Authors · 2025-11-24
> **Rebuttal by Authors (part 3 of 4)**
>
> > **Q1:** How does GENTS scale computationally with increasing sequence length and variable dimensions? What is the theoretical complexity compared to task-specific baselines?
>
> We thank the reviewer for the question on the computational scaling of GenTS with respect to sequence length and variable dimensions, and how this compares to task-specific baselines.
>
> For fair comparison, we focus on the time-series backbones. Multimodal models like GenTS contain both a time-series backbone and additional text modules, while most baselines in our experiments are unimodal and only have the backbone. To avoid mixing architectural details that are not directly comparable, we therefore analyze only the backbone complexity.
>
> Specifically, let the input time series have length $L$, patch size $P$, and hidden dimension $D$. We define $N=L/P$ for the number of time-series patches. For clarity, in the following discussion we omit the number of layers and focus on how the backbone complexity scales with the sequence length and the number of variables. The full expressions can be recovered by multiplying the stated terms by the corresponding layer count $J$ for each model.
>
> (i) GenTS has standard Transformer complexity $O(N^2D)$ and extends linearly to multivariate series.
>
> - Univariate. Firstly, we clarify that the current experiments in our paper are conducted on univariate time series. Under this setting, each layer applies self-attention and a feed-forward block on the $N$ patches, leading to the per-layer cost $O(N^2D + ND^2)$, whose dominant term in practice is $O(N^2D)$. Ignoring constant factors and the layer count, the training complexity is $O(N^2D)$.
> - Multivariate. If we extend GenTS to multivariate time series with $K$ variables, we can follow the widely used channel-independent strategy which is also adopted in prior work such as PatchTST[7]. In that case, the backbone complexity simply multiplies by $K$, giving $O(KN^2D)$.
>
> So GenTS scales quadratically with the patched sequence length $N$ and linearly with the number of variables $K$.
>
> (ii) Linear and CNN baselines have $O(KLD)$ backbones.
> - Linear models such as DLinear [8] use per-channel linear layers over the original sequence of length $L$. Their backbone training cost is therefore linear in the sequence length and the number of variables. It can be written as $O(KLD)$.
> - CNN-based models such as TimesNet [9] apply convolutions with fixed kernel sizes and channel widths. Their dominant backbone cost also scales approximately linearly in $L$ and $K$ which is $O(KLD)$.
>
> (iii) Other Transformer baselines such as Autoformer [10] and Transformer-based diffusion baselines such as Diffusion-TS [11] match the $O(KN^2D)$ backbone order of GenTS.
> - If they use patching, their training complexity is also $O(KN^2D)$ for the multivariate channel-independent setting.
> - Otherwise, the cost is $O(KL^2D)$ without patching.
>
> [7] A Time Series is Worth 64 Words: Long-term Forecasting with Transformers. ICLR 2023.
>
> [8] Are Transformers Effective for Time Series Forecasting? AAAI 2023.
>
> [9] TimesNet: Temporal 2D-Variation Modeling for General Time Series Analysis. ICLR 2023.
>
> [10] Autoformer: Decomposition Transformers with Auto-Correlation for Long-Term Series Forecasting. NeurIPS 2021.
>
> [11] Diffusion-TS: Interpretable Diffusion for General Time Series Generation. ICLR 2024.

---

> ### Author Response · Authors · 2025-11-24
> **Rebuttal by Authors (part 4 of 4)**
>
> > **Q2:** What are the failure modes of the domain-agnostic captioning approach? Can it capture complex patterns like multi-scale anomalies or non-stationary transitions that lack clear statistical signatures?
>
> We thank the reviewer for raising this meaningful question about the failure modes of our domain-agnostic captioning.
>
> We acknowledge that, although our domain-agnostic captioning is effective in most cases, it does exhibit several identifiable failure modes.
> - Complex feature combinations. While we capture multi-scale information through "Overall/Local Trend" and anomaly signatures through "Kurtosis/Number of Peaks", more complex combinations of features (such as multi-scale anomalies) are difficult to describe accurately.
> - High-frequency signals and diverse fine-grained shapes. Most of our attributes emphasize low-frequency characteristics, and the local shapelets are limited to fixed types, making it challenging to capture high-frequency or diverse local structures.
>
> These failure modes arise primarily from constraints on textual caption length and limitations of external tools like tsfresh [12], which are applied to extract the attributes from time series.
> - Caption length and domain-agnostic constraints. To maintain domain-agnostic applicability and efficiency, captions are intentionally kept short and descriptive, summarizing only key attributes. This design naturally omits very fine-grained or high-frequency details that would require longer or domain-specific explanations.
> - Dependence on the attribute set of external tools. The expressiveness of the captions is bounded by the attribute set provided by external tools. Although the attribute set is broad, it remains predefined, and the captioning pipeline inherits the blind spots of these tools.
>
> We discuss potential methods to mitigate the impact of these failure modes.
> - Enriching the attribute set with more complex attribute combinations. Extending the attribute toolbox would enable the model to capture more complex time-series patterns, such as multi-scale anomalies and non-stationary transitions.
> - Optional LLM-based refinement for richer descriptions. LLMs can be used to refine or expand the attribute-based captions into more expressive and detailed descriptions when needed.
>
> We greatly appreciate the reviewer's insightful question. Exploring more comprehensive and efficient approaches for time series caption is indeed a meaningful direction, and we hope our method provides useful inspiration of multimodal dataset construction for future research.
>
> [12] tsfresh: Time Series Feature extraction based on scalable hypothesis tests. Neurocomputing, 2018.
>
> > **Q3:** How does the unified training objective balance potentially conflicting optimization requirements between forecasting and generation tasks? The paper mentions λF:λG=2:1 works well, but provides no theoretical justification.
>
> We thank the reviewer for these questions.
>
> We believe that forecasting and generation are not fundamentally conflicting in our framework, and the choice of loss weights is informed by both the relative scales of the two losses and an empirical sensitivity study.
>
> In our framework, forecasting and generation are not contradictory; instead, they are complementary to some extent.
> - Both forecasting and generation are forms of conditional probabilistic modeling. They learn to model $P(\text{target}\mid\text{condition})$. Forecasting conditions on text to predict future series, while generation uses similar mechanisms but with different masks and prompts.
> - Since both tasks optimize related distributions rather than opposing objectives, they can reinforce each other. Forecasting encourages precise modeling of temporal dependencies, whereas generation promotes capturing global structure under textual and structural conditions.
>
> The choice of loss weights is motivated by the relative loss scales and practical evaluation priorities.
> - In our experiments, the raw forecasting loss and generation loss have average magnitudes roughly in the ratio $L\_F:L\_G = 1:2$, where the forecasting loss and generation loss are defined in Sec.3.2 of our paper. Using this weighting strategy gives both tasks comparable influence during training and prevents either loss from dominating the gradients due to scale differences.
> - Conceptually, forecasting is evaluated with strict metrics such as MSE, which are more sensitive to small prediction errors compared to generation metrics. Assigning a higher weight to forecasting prioritizes the learning of precise temporal dependencies.
> - Empirically, our primary evidence comes from the sensitivity analysis in Fig.4 of our paper. This experiment shows that the model's performance is highly robust across a wide range of loss weight ratios. The 2:1 ratio is not a fragile optimum, in contrast, it consistently provides balanced, strong performance across multiple datasets.

---

> > ### Comment · Reviewer_woXw · 2025-11-27
> > **Official Comment by Reviewer woXw**
> >
> > Thank you for your detailed and comprehensive reply. You have resolved most of my concerns. However, regarding the model efficiency, not limited to theoretical analysis, you can post detailed results such as training time, memory usage, and compare them with relevant baselines. This will be more convincing. Therefore, I will temporarily maintain my score.

---

> ### Author Response · Authors · 2025-11-27
> **About remaining question**
>
> Thank you for recognizing the contributions of our work and the improvements made during the rebuttal. In this response, we focus on addressing the remaining concern regarding training time and memory usage.
>
> Per the reviewer's suggestions, we conduct a comparison of memory usage and training time. We compare our method (GenTS) with a unimodal diffusion model (DiffusionTS [1]) and a multimodal non-diffusion model (IATSF [2]).
> - Experiment setting. We conduct the experiment on the Weather dataset, with a historical length of 36 and forecasting horizons of {36, 72, 108}. The batch size is set to 1024 and the number of epochs to 100 for all methods. All experiments are run on a single NVIDIA A40 GPU. For our method (GenTS), we include both the original multimodal implementation and the unimodal variant (i.e., without the text encoder and textual information) to examine the impact of multimodal modeling.
> - Experiment results. As shown in Table 1-2 below, the results can be summarized as follows:
>     - **The memory requirements are modest, and our model is compatible with most types of GPUs.** Compared with the unimodal diffusion model (DiffusionTS), the additional memory usage of GenTS primarily comes from the parameters, activations, and gradients of the text encoder. Nevertheless, our GPU memory consumption remains significantly lower than that of the multimodal model (IATSF). Importantly, the ~7500 MB memory requirement during training can be satisfied by most commonly used GPUs.
>     - **The training time is acceptable.** The training time of GenTS, DiffusionTS, and IATSF is of the same order of magnitude, which aligns with the complexity analysis provided in our response to W2. Compared with the unimodal diffusion model (DiffusionTS), the extra training time of GenTS is mainly due to multimodal modeling, which is a reasonable trade-off for improved performance.
>
> #### Table 1. Peak memory usage of different methods on Weather dataset across different forecasting horizons. The unit of the memory usage is Megabyte.
> |Horizon|GenTS|GenTS w/o text|DiffusionTS [1] |IATSF [2] |
> |:--:|:--:|:--:|:--:|:--:|
> |36|7273|1988|3704|23284|
> |72|7593|2514|6454|23284|
> |108|7860|3154|9602|23284|
>
> #### Table 2. Training time of different methods on Weather dataset across different forecasting horizons. The unit of the training time is minute.
> |Horizon|GenTS|GenTS w/o text|DiffusionTS [1] |IATSF [2] |
> |:--:|:--:|:--:|:--:|:--:|
> |36|40|4.4|11.2|45|
> |72|41|5.7|17.8|45.6|
> |108|42|7.2|27.3|46.7|
>
> In addition to the quantifiable training time and memory usage, our unified model also offers potential benefits in terms of model building and parameter efficiency, which are hard to directly compare.
> - Time. A unified model can address multiple tasks with a single architecture and training pipeline, reducing the effort required to design and train separate models for different tasks.
> - Space. A unified model shares the same architecture and parameters across tasks, thereby saving GPU memory during training and reducing model storage requirements.
>
> We hope the above analysis resolves the reviewer's question regarding training time and memory usage, and we look forward to receiving your feedback or any further specific questions.
>
> [1] Diffusion-TS: Interpretable Diffusion for General Time Series Generation. ICLR 2024.
>
> [2] Intervention-Aware Forecasting: Breaking Historical Limits from a System Perspective. 2025.

---

### Official Review · Reviewer_7hBc · 2025-10-27

**Soundness:** 2
**Presentation:** 2
**Contribution:** 2
**Rating:** 2
**Confidence:** 4

**Summary:**

This paper introduces GENTS, a novel unified generative model specifically designed for multimodal time series analysis, which seeks to overcome the limitations of existing approaches that often only treat external modalities, such as textual descriptions, as supplementary features. The core contribution is a cohesive framework that explicitly models the joint distribution between the time series and auxiliary data to enhance generalization and performance across various prediction and editing tasks. The authors demonstrate that GENTS achieves significant performance improvements over strong foundation models and dedicated diffusion baselines on several distinct time series datasets.

**Strengths:**

1. The motivation behind this work is both novel and valuable, advocating for a shift from treating auxiliary modalities merely as supplementary features to explicitly modeling the joint distribution between the time series and the external data for richer analysis.

2. The proposed GENTS methodology appears sound and offers a reasonable, unified generative approach to incorporate diverse multimodal data, which is essential for advancing the generality of time series models.

3. The experimental evaluation is relatively comprehensive, showcasing the model's superior performance across multiple disparate time series tasks against several competitive and recently proposed foundation and diffusion baselines.

**Weaknesses:**

The primary confusion lies in the clarity of the experimental setup, which fundamentally impacts the assessment of novelty and fairness. Specifically, it remains ambiguous whether GENTS is implemented as a multi-task framework (trained and evaluated independently on each task's dataset) or if it functions as a single, large pre-trained foundation model evaluated via zero-shot transfer on downstream tasks. If this work falls into the former category (separate training per task), its technical novelty is somewhat limited; however, if the authors pursued the latter (unified pre-training followed by zero-shot evaluation), the comparison to task-specific baselines is likely unfair as the baselines were not optimized under equivalent pre-training data conditions.

**Questions:**

Please refer to Weaknesses.

---

> ### Author Response · Authors · 2025-11-24
> **Rebuttal by Authors (part 1 of 2)**
>
> > **W1:** The primary confusion lies in the clarity of the experimental setup, which fundamentally impacts the assessment of novelty and fairness. Specifically, it remains ambiguous whether GENTS is implemented as a multi-task framework (trained and evaluated independently on each task's dataset) or if it functions as a single, large pre-trained foundation model evaluated via zero-shot transfer on downstream tasks. If this work falls into the former category (separate training per task), its technical novelty is somewhat limited; however, if the authors pursued the latter (unified pre-training followed by zero-shot evaluation), the comparison to task-specific baselines is likely unfair as the baselines were not optimized under equivalent pre-training data conditions.
>
> We thank the reviewer for the thoughtful comments regarding our training paradigm and its implications for novelty and fairness. Below, we provide clarifications on the training setup, technical novelty, and evaluation fairness, followed by a summary of the corresponding revisions made to the paper to address these concerns.
>
> > W1.1: The primary confusion lies in the clarity of the experimental setup, which fundamentally impacts the assessment of novelty and fairness. Specifically, it remains ambiguous whether GENTS is implemented as a multi-task framework (trained and evaluated independently on each task's dataset) or if it functions as a single, large pre-trained foundation model evaluated via zero-shot transfer on downstream tasks.
>
> **Clarification of the training setup.**
> First, we clarify that GenTS is a dataset-specific unified model.
> - Unified multi-task per dataset: GenTS is implemented as a unified multi-task framework trained separately on each dataset, rather than as a single pre-trained model. For each dataset in Table 2 of our paper, we train one GenTS model on all relevant tasks using that dataset's training split and evaluate it on the corresponding test split.
> - Paper revision per reviewer's suggestions: To avoid any ambiguity, we have revised Sec. 4.1 to explicitly state that GenTS is trained independently per dataset and that the multi-task setup is constrained within each dataset.
>
> **Rationale for unified multi-task training within each dataset.**
> We would also like to clarify the motivation behind our design choice and explain why the pre-training paradigm suggested by the reviewer is orthogonal to the scope of this work.
> - Unified multi-task learning as a foundational capability: Multi-task learning is a core capability for modern machine learning models and serves as an essential step toward building unified foundation models. In the context of time-series modeling, jointly learning heterogeneous generative tasks remains challenging and relatively unexplored. Our work takes a first step in this direction by unifying forecasting, generation, and editing within a single generative framework, providing a principled basis for future research on more comprehensive foundation models.
> - Orthogonality to large-scale pre-training: The large-scale pre-training paradigm recommended by the reviewer (such as multi-domain dataset construction, high-efficiency training pipelines, and zero-shot generalization) is complementary to, but distinct from, the primary focus of our work. Our contribution lies in unifying the model architecture and task formulation across diverse time-series tasks within a single model. While GenTS can naturally benefit from future incorporation of large-scale pre-training techniques, fully developing such a paradigm goes beyond the current scope. We therefore view it as a promising direction for future exploration.

---

> ### Author Response · Authors · 2025-11-24
> **Rebuttal by Authors (part 2 of 2)**
>
> > W1.2: If this work falls into the former category (separate training per task), its technical novelty is somewhat limited;
>
> Our contribution centers on introducing a unified masked-conditional generation paradigm for multi-task and multimodal time-series modeling.
> - We propose a masked-conditional generation paradigm that unifies input/output formats and optimization objectives across forecasting, generation, and editing, thereby enabling flexible adaptation within a single architecture.
> - To the best of our knowledge, this is the first generative framework to jointly address multi-task and multimodal time series modeling within a unified model. This unified design reduces development cost and avoids the complexity associated with maintaining multiple task-specific models.
> - Our approach also provides a solid foundation for future extensions to additional tasks and modalities, demonstrating strong potential for expansion to multimodal outputs and broader applications.
>
> To the best of our knowledge, there is no widely recognized multimodal foundation model in the time series domain, largely due to challenges such as data scarcity and the complexity of modeling multimodal distributions. Our contribution and novelty are orthogonal to the concept of a foundation model, as we focus on unifying different tasks rather than generalizing across different domains. Nevertheless, we believe our work provides both a conceptual foundation and a technical direction toward building a multimodal unified time series foundation model in the future.
>
> > W1.3: however, if the authors pursued the latter (unified pre-training followed by zero-shot evaluation), the comparison to task-specific baselines is likely unfair as the baselines were not optimized under equivalent pre-training data conditions
>
> Sorry for any caused confusion about the evaluation setup.
>
> The comparisons to baselines are fair because GenTS and all non-foundation model baselines are trained and evaluated under identical conditions.
>
> - Since our GenTS is trained separately on each dataset, both GenTS and the task-specific baselines use the same training and evaluation splits, ensuring fairness in all results reported in Table 2 of our paper.
> - The zero-shot experiment in Table 4 of our paper is presented solely as an additional cross-dataset analysis to assess generalization enabled by domain-agnostic captions. It is not used for direct comparison with dataset-specific baselines.
>
> **To address the reviewer's concern of Weakness 1 regarding clarity, we have revised our paper accordingly**. Specifically, we have revised Sec.4.1 to explicitly describe GenTS as a unified multi-task model trained separately for each dataset, and we have updated Sec.4.4 to clearly label Table 4 as a zero-shot cross-dataset analysis focused on domain-agnostic captions. We believe these revisions make the training paradigm, novelty, and fairness of our comparisons fully transparent.

---

### Official Review · Reviewer_6ZFC · 2025-10-29

**Soundness:** 3
**Presentation:** 3
**Contribution:** 2
**Rating:** 2
**Confidence:** 4

**Summary:**

This paper proposes GenTS, a unified generative framework based on diffusion models for multimodal time series analysis. The core idea is to model time series data and textual descriptions together, with the ability to diverse tasks— forecasting, unconditional generation, and editing—within a single masked-conditional generation paradigm.

Another contribution is a domain-agnostic time series captioning method that automatically generates textual descriptions from statistical and morphological attributes because of the lack of real data.

**Strengths:**

In general, this paper is comprehensive and try to bring traditional time series tasks to next level. Also, this paper did multiple experiments to prove the model's capabilities in forecasting, editing, and generation.

**Weaknesses:**

1.The paper should clarify what is unified modeling in the time series case. Because the general concept of unified modelling should be training with multiple tasks. But in this paper, although there are 3 tasks, the editing task is training-free. So in essense, this paper only targets in conditional generation and unconditional generation, which is very common in controllable generation, and ** should not be named unified modelling.**  Some real unified time series modelling papers include TimeDiT, UrbanDiT.

2. Unclear dataset description in editing task. This paper used real-world dataset with text to test editing tasks, but the experimental details are unclear, for example, for TimeMMD-Env, how to define the orignal time series?

3. Also, there should be some case study to visualize the capability of edting.

4. In the model architechture, the text is incorporated by adaLN, however, adaLN only suitable for global text instead of detailed text, for the tasks in this paper, cross-attention to incorporated the text should be more appropriate. You can see the GenTron and HunyuanDiT paper to know why cross-attention should be used. Also, cross-attention experiments should be added.

5. Why not use the flow-matching, because in generation area, flow matching has supassed diffusion.

**Questions:**

1. On the unified modeling: Can clarify the defination or change the name?

2. On the experimental setup for editing: Can you detail the precise process for creating the 'new textual guidance' used for editing the real-world datasets like TimeMMD-Env?

3. On the model architecture choice: What was the specific reasoning behind using an AdaLN mechanism for text conditioning instead of a cross-attention layer, which is commonly used for detailed textual control?

4. On the choice of generative backbone: Was there a specific reason for choosing a diffusion model as the backbone over more recent alternatives like Flow Matching?

---

> ### Author Response · Authors · 2025-11-24
> **Rebuttal by Authors (part 1 of 4)**
>
> > **W1:** The paper should clarify what is unified modeling in the time series case. Because the general concept of unified modelling should be training with multiple tasks. But in this paper, although there are 3 tasks, the editing task is training-free. So in essense, this paper only targets in conditional generation and unconditional generation, which is very common in controllable generation, and **should not be named unified modelling.** Some real unified time series modelling papers include TimeDiT, UrbanDiT.
>
> > W1.1 The paper should clarify what is unified modeling in the time series case. Because the general concept of unified modelling should be training with multiple tasks. But in this paper, although there are 3 tasks, the editing task is training-free.
>
> We thank the reviewer for raising this important question about what we mean by unified modeling. In brief, our definition of "unified" focuses on a single model capable of solving diverse tasks.
>
> As stated in the introduction of our paper, our goal is to establish a unified generative paradigm in which one model can address multiple time series tasks. Our notion of unification emphasizes a single model, a unified training process, and the ability to solve various tasks without additional adaptation.
>
> The fact that the editing task is training-free is not a limitation; rather, it demonstrates that our model has learned a robust and generalizable representation, enabling editing capabilities to emerge naturally through joint training. In unified modeling, not all downstream tasks need to have their own explicit training objective.
>
> > W1.2 So in essense, this paper only targets in conditional generation and unconditional generation, which is very common in controllable generation, and should not be named unified modelling.
>
> We thank the reviewer for this comment. However, we would like to clarify that our framework does not perform unconditional generation. All tasks within our framework are inherently conditional.
>
> Our unification is achieved through a masked conditional generation paradigm, as illustrated in Fig.1 of our paper.
> - Forecasting task is formulated as generating a future series, conditioned on the past series and text.
> - Generation task is formulated as generating an entire series, conditioned only on text.
> - Editing task is formulated as generating a target series, conditioned on text and a source series.
>
> The reviewer's perspective might oversimplify the distinction between time series forecasting and generation.
> - In the time series community, forecasting and generation are two independently developing subfields.
> - Treating both tasks merely as "controllable generation" overlooks the fundamental differences in their training objectives, evaluation metrics, and the model architectures traditionally used.
>
> Our unified approach is also empirically synergistic.
> - The ablation study in Finding 3 of our paper provides direct evidence that our unification is more than the sum of its parts: jointly training the model on the generation task actively improves forecasting performance.
>
> Therefore, "unified modeling" accurately characterizes our contribution, which lies in integrating several distinct and challenging time series tasks within a single modeling framework. We have added a more explicit explanation of the term "unified" in Sec.1 of the revised version of our paper.

---

> ### Author Response · Authors · 2025-11-24
> **Rebuttal by Authors (part 2 of 4)**
>
> > W1.3 Some real unified time series modelling papers include TimeDiT, UrbanDiT.
>
> We thank the reviewer for pointing out related work such as TimeDiT [1] and UrbanDiT [2], and for prompting us to further clarify our use of the term unified modeling.
>
> We distinguish between foundation models and unified models.
> - A foundation model is trained on multi-domain datasets, with the primary goal of achieving zero-shot generalization to new domains.
> - A unified model is designed to handle multiple tasks within a single architecture and training process.
>
> Under these definitions, TimeDiT and UrbanDiT can be viewed as both foundation and unified models, whereas our GenTS is a unified model for multiple tasks within a single domain. We do not claim GenTS to be a foundation model.
>
> Nevertheless, GenTS is a unified model in a multimodal setting (time series + text), while TimeDiT and UrbanDiT operate in a unimodal setting (time series only). We believe the multiple modalities introduce both benefits and unique challenges.
> - Benefits. Additional modalities provide richer information for all tasks. In forecasting, the model can make more accurate predictions when guided by textual descriptions. In generation, the model gains more refined controllability through text. As shown in Finding 2, the performance of both forecasting and generation drops noticeably when text is removed.
> - Challenges. The model must capture a more complex multimodal joint distribution, requiring careful alignment across modalities. The scarcity of paired multimodal data further limits broader applicability. We discuss this in the Sec. 1 of our paper.
>
> Our current focus is on developing a robust unified multimodal framework within a single domain before scaling toward a full foundation model. Nonetheless, we believe our work provides meaningful insights for future research on multimodal unified foundation models. Thank you again for your valuable comments.
>
> [1] TimeDiT: General-purpose Diffusion Transformers for Time Series Foundation Model. 2024.
>
> [2] Diffusion Transformers as Open-World Spatiotemporal Foundation Models. NeurIPS 2025.
>
> > **W2:** Unclear dataset description in editing task. This paper used real-world dataset with text to test editing tasks, but the experimental details are unclear, for example, for TimeMMD-Env, how to define the orignal time series?
>
> We thank the reviewer for pointing out that the experimental setup for the editing task was unclear. Per your suggestion, we have added a clear description of this protocol to Appendix C.3 in the revised version.
>
> We follow TEdit [3] to construct the editing pairs, but use unstructured text instead of fixed attributes. Specifically, TEdit defines editing as transforming a source time series into a target style, where both are associated with structured attributes. Our construction pipelines are as follows.
> - Let $\mathcal{D}$ denote a dataset of time–text pairs, and $\mathcal{D} = \{(x\_i, c\_i)\}\_{i=1}^N$, where $x\_i$ is a time series and $c\_i$ is its associated textual description. We split $\mathcal{D}$ into training and test subsets, and use only the test split $\mathcal{D}\_{\text{test}} \subset \mathcal{D}$ for the editing experiments.
> - For each editing instance on a given dataset, we construct a source–target pair by sampling two distinct elements from $\mathcal{D}\_{\text{test}}$. Formally, $(x\_{\text{src}}, c\_{\text{src}}), \ (x\_{\text{tgt}}, c\_{\text{tgt}}) \sim \mathcal{D}\_{\text{test}}, \quad (x\_{\text{src}}, c\_{\text{src}}) \neq (x\_{\text{tgt}}, c\_{\text{tgt}})$. The target of the editing is to modify the $x\_{\text{src}}$ given the target caption $c\_{\text{tgt}}$.
>
> We add a clearer, step-by-step explanation of the editing protocol in Appendix C.3 of the revised version. We hope this clarification resolves the ambiguity around the editing data construction and makes the experimental setup easier to reproduce.
>
> [3] TEdit: Towards Editing Time Series. NeurIPS 2024.
>
> > **W3:** Also, there should be some case study to visualize the capability of edting.
>
> We thank the reviewer for this suggestion.
>
> We have already provided visual case studies in Appendix G.2 of our paper, covering forecasting, generation, and editing. These results demonstrate that GenTS is capable of producing reasonable time series across diverse datasets and tasks.
>
> To further illustrate the editing capabilities of GenTS, we have added additional editing examples in Fig. 8, which clearly show that GenTS can successfully transform the original series to align with the new conditioning information.

---

> ### Author Response · Authors · 2025-11-24
> **Rebuttal by Authors (part 3 of 4)**
>
> > **W4:** In the model architechture, the text is incorporated by adaLN, however, adaLN only suitable for global text instead of detailed text, for the tasks in this paper, cross-attention to incorporated the text should be more appropriate. You can see the GenTron and HunyuanDiT paper to know why cross-attention should be used. Also, cross-attention experiments should be added.
>
> We thank the reviewer for the thoughtful comments.
>
> We believe that conclusions drawn from cross-attention in image/video diffusion models are not directly transferable to our multimodal time series setting, as time series captions are far less information-dense than vision prompts. Moreover, our additional experiments show that cross-attention not only degrades forecasting performance but also fails to provide consistent improvements in generation, while simultaneously increasing computational overhead.
>
> Cross-attention designs that succeed in vision diffusion models may not work well for time series. The effectiveness of cross-attention in GenTron [4] and HunyuanDiT [5] is demonstrated in high-resolution text-to-image and text-to-video generation, where models are conditioned on long, information-rich textual prompts. In contrast, time series captions are typically shorter, more structured, and contain less semantic variability. As a result, token-level cross-attention introduces the risk of overfitting to specific tokens instead of capturing the numerical and global temporal patterns essential for time series.
>
> Our experiments show that cross-attention underperforms the vanilla GenTS in forecasting and does not provide consistent improvements in generation. As summarized in Table 1 below, forecasting accuracy decreases on both datasets, and only the generation fidelity on the Traffic dataset exhibits a slight improvement.
>
> There is currently no universally agreed-upon optimal method for incorporating conditions into diffusion models in the context of multimodal time series, but we believe this remains a promising direction for future exploration. We thank the reviewer again for the helpful suggestions.
>
> [4] GenTron: Diffusion Transformers for Image and Video Generation. CVPR 2024.
>
> [5] Hunyuan-DiT : A Powerful Multi-Resolution Diffusion Transformer with Fine-Grained Chinese Understanding. 2024.
>
> #### Table 1. The impact of cross-attention. $\uparrow$ ($\downarrow$) means the higher (lower), the better.
>
> |Task|Forecasting|Forecasting|Forecasting|Forecasting|Generation|Generation|Generation|Geneartion|
> |:--:|:--:|:--:|:--:|:--:|:--:|:--:|:--:|:--:|
> |Dataset|HongKong|HongKong|Traffic|Traffic|HongKong|HongKong|Traffic|Traffic|
> |Metric|$\downarrow$MAE|$\downarrow$MSE|$\downarrow$MAE|$\downarrow$MSE|$\downarrow$JFTSD|$\uparrow$CTTP|$\downarrow$JFTSD|$\uparrow$CTTP|
> |Cross attention|0.848|1.284|0.502|0.500|25.174|137.265|163.143|44.902|
> |GenTS (ours)|0.743|1.003|0.456|0.420|23.516|140.052|178.937|44.942|
>
> > **W5:** Why not use the flow-matching, because in generation area, flow matching has supassed diffusion.
>
> We thank the reviewer for this question.
>
> We clarify that our primary goal is to introduce a novel unification paradigm, rather than proposing a new generation mechanism. To validate this core contribution, we build upon the most mature, robust, and well-understood generative framework, denoising diffusion models. This choice allows us to attribute performance differences primarily to the proposed unified modeling paradigm.
>
> Using diffusion models keeps our comparisons fair and isolates the effect of our unified framework from changes in the generative backbone. The generative baselines we compare against are diffusion-based models. By adopting the same generative mechanism, we ensure that any observed improvements can be more directly attributed to our unified multimodal framework, rather than to a new or different generation technique.
>
> We view flow matching as a promising alternative within our framework. However, applying flow matching to time series introduces its own challenges, such as low signal-to-noise ratios and the difficulty of designing stable vector fields for irregular temporal dynamics. These are interesting avenues for future research.

---

> ### Author Response · Authors · 2025-11-24
> **Rebuttal by Authors (part 4 of 4)**
>
> > **Q1:** On the unified modeling: Can clarify the defination or change the name?
>
> We thank the reviewer for the follow-up question on the unified modeling terminology.
>
> As we clarified in the responses to **W1.1–W1.3**, our intention is not to misuse the term "unified", but to use it in a precise sense: a single masked-conditional multimodal generative model that handles forecasting, generation, and editing within one backbone and one training process. We agree that this definition should be stated more explicitly, and we have refined it in the latest version of our paper. We will retain the notion of unified modeling, as it accurately reflects our contribution.
>
> > **Q2:** On the experimental setup for editing: Can you detail the precise process for creating the 'new textual guidance' used for editing the real-world datasets like TimeMMD-Env?
>
> We thank the reviewer for this follow-up question regarding the editing setup. As detailed in our response to **W2** and in the revised Appendix C.3 of our paper, the "new textual guidance" is randomly sampled from other samples within the same dataset.
>
> In all editing experiments, including TimeMMD-Env, the new textual guidance is simply the caption associated with a different target time series from the same test split. In other words, we treat the descriptive text of another real sample as the editing instruction.
>
> As a result, the guidance always reflects realistic descriptions from the same domain, and the editing task corresponds to transforming an original series so that it aligns with the semantics of a different caption. The full construction details have been clarified in **W2** and will be precisely documented in Appendix C.3 in the revised version.
>
> > **Q3:** On the model architecture choice: What was the specific reasoning behind using an AdaLN mechanism for text conditioning instead of a cross-attention layer, which is commonly used for detailed textual control?
>
> We thank the reviewer for the follow-up question regarding the model architecture.
>
> As clarified in our response to **W4**, our architectural choice is motivated by both empirical evidence and the requirements of our multimodal time series setting. Moreover, the comparison with the cross-attention confirms that GenTS already provides strong text-controlled generation while offering better efficiency.
>
> > **Q4:** On the choice of generative backbone: Was there a specific reason for choosing a diffusion model as the backbone over more recent alternatives like Flow Matching?
>
> We thank the reviewer for the follow-up question about our choice of generative backbone.
>
> As discussed in our response to **W5**, our decision to use a diffusion model is primarily motivated by the focus of this paper on unified multimodal time series modeling, rather than on exploring the advanced generative mechanism. Using diffusion also enables fair comparison with diffusion-based baselines, while allowing us to treat flow matching as a promising yet orthogonal direction for future research.

---

### Official Review · Reviewer_1vhp · 2025-11-01

**Soundness:** 2
**Presentation:** 3
**Contribution:** 2
**Rating:** 6
**Confidence:** 3

**Summary:**

This paper proposes GENTS, a diffusion-based framework that learns the joint distribution of time series and text, unifying forecasting, generation, and editing under a masked conditional generation paradigm.

**Strengths:**

Integrates forecasting, generation, and editing into a single framework.

Shows significant gains over TimeWeaver and TEdit on generation and editing; for forecasting across multiple datasets, performance is competitive with or better than strong baselines.

Clear and easy to follow.

**Weaknesses:**

Conceptually similar to conditional time-series generation (e.g., TimeWeaver) and text-guided editing (e.g., TEdit); those works already introduced J-FTSD/RaTS. The main novelty here is the unified packaging and the captioner.


Baselines include Chronos and Moirai, but the area is moving fast; consider adding newer models on GIFT-Eval, such as Chronos-2, Moirai-2, and TimesFM-2.5.


The caption pipeline uses tsfresh attributes templated into text; this ensures cross-domain applicability but limits expressiveness and interpretability.


GENTS is noticeably slower than non-diffusion baselines.


A uniform setup is used across datasets (700 epochs, bs=1024, lr=1e-3) with 3 seeds on an A40; fairness to all baselines is unclear.


In Table 2 (Weather), J-FTSD shows a large scale gap (GENTS = 1.320 vs. ~290 for two strong baselines); please further explain this anomalous advantage.


Beyond automatic metrics, consider adding a small human preference study (fidelity and instruction alignment) to support the conclusions.

**Questions:**

Please check weakness

---

> ### Author Response · Authors · 2025-11-24
> **Rebuttal by Authors (part 1 of 3)**
>
> > **W1:** Conceptually similar to conditional time-series generation (e.g., TimeWeaver) and text-guided editing (e.g., TEdit); those works already introduced J-FTSD/RaTS. The main novelty here is the unified packaging and the captioner.
>
> We thank the reviewer for pointing out the related works TimeWeaver [1] and TEdit [2].
>
> TimeWeaver and TEdit are representative works for conditional generation and editing, respectively. In this work, we draw on the evaluation metrics used in these papers for assessing the corresponding tasks and include them as baselines for comparison. Nevertheless, they differ from GenTS (our method) in several important ways.
> - *GenTS is a unified time series model*. GenTS considers not only generation and editing but also forecasting. Forecasting is fundamentally different from the other two tasks, especially in terms of evaluation metrics. Balancing the performance of all three tasks within a single unified model presents a significant challenge.
> - *GenTS is text-guided, whereas TimeWeaver and TEdit are not*. Both TimeWeaver and TEdit rely on structured metadata as their conditions for generation and editing, whereas GenTS uses textual descriptions. Text provides richer and more fine-grained semantics, enabling more precise generation and editing. As shown in Table 2 of our paper, GenTS achieves superior performance in both generation and editing compared with these two baselines.
>
> [1] Time Weaver: A Conditional Time Series Generation Model. ICML 2024.
>
> [2] Towards Editing Time Series. NeurIPS 2024.
>
> > **W2:** Baselines include Chronos and Moirai, but the area is moving fast; consider adding newer models on GIFT-Eval, such as Chronos-2, Moirai-2, and TimesFM-2.5.
>
> We thank the reviewer for pointing out the recent progress of time-series foundation models on GIFT-Eval and for suggesting Chronos-2, Moirai-2, and TimesFM-2.5 as additional baselines.
>
> Our existing baselines already provide *strong and representative coverage* of both classical and foundation time-series models. GenTS is compared against a diverse set of unimodal and multimodal baselines, the foundation models, as well as multimodal models. These baselines cover MLP-, CNN-, Transformer-, diffusion-based, and foundation-model families, and they are widely adopted in recent time-series benchmarks. The main-paper results in Table 2, with more detailed results in Table 7–8, show that GenTS already achieves strong performance compared with these representative models across multiple datasets and tasks, supporting the validity of our empirical conclusions.
>
> The mentioned models, i.e., Chronos-2, Moirai-2, and TimesFM-2.5, are the latest time-series foundation models that appeared following their previous version Chronos, Moirai, and TimesFM, and they were released only after we had finalized our experimental pipeline and submitted the paper. Consequently, they were not available to us during the initial experimental phase, which explains their absence as baselines in the submitted version.
>
> Per your suggestion, we have now added Chronos-2, Moirai-2, and TimesFM-2.5 to our evaluation on ETTm1, HongKong, and Weather datasets. The results are shown in Table 1 below, confirming that our main empirical conclusions remain unchanged. When these new baselines are considered with the results already reported for GenTS and existing models, we observe that the relative positioning of GenTS in the overall ranking remains essentially unchanged. In particular, the additional comparisons do not alter our key take-away that *GenTS is a strong and competitive unified generative framework for time-series forecasting, generation, and editing*.
>
> #### Table 1. Compare with Chronos2, Moirai2, TimesFM2.5. $\downarrow$ means the lower the better.
>
> |Method|ETTm1 $\downarrow$MAE|ETTm1 $\downarrow$MSE|HongKong $\downarrow$MAE|HongKong $\downarrow$MSE|Weather $\downarrow$MAE|Weather $\downarrow$MSE|
> |:---:|:--:|:--:|:--:|:--:|:--:|:--:|
> |Chronos2 (zero-shot)|0.866|1.681|0.822|1.222|0.303|0.249|
> |Moirai2 (zero-shot)|0.809|1.501|0.820|1.205|0.316|0.236|
> |TimesFM2.5 (zero-shot)|0.779|1.326|0.799|1.190|0.296|0.227|
> |GenTS (ours)|0.722|1.168|0.743|1.003|0.221|0.139|

---

> ### Author Response · Authors · 2025-11-24
> **Rebuttal by Authors (part 2 of 3)**
>
> > **W3:** The caption pipeline uses tsfresh attributes templated into text; this ensures cross-domain applicability but limits expressiveness and interpretability.
>
> Thank you for the suggestions.
>
> The primary goal of our caption design is to obtain domain-agnostic, attribute-grounded descriptions. The motivation behind templating tsfresh attributes into text is to ensure that each caption sentence corresponds clearly to a specific time series property. By grounding the text in tsfresh-style features, the same captioning pipeline can be applied consistently across different domains without requiring manual engineering or domain-specific templates.
>
> The attribute-based captions are sufficiently expressive and interpretable in practice, as demonstrated by our experiments.
> - Expressiveness in forecasting. Fig.3 in our paper shows that all attributes described in the captions contribute to performance improvement, indicating that the combined attribute descriptions capture rich predictive information.
> - Interpretability. Tab.6 in our paper shows that human subjects systematically prefer our domain-agnostic captions over GPT-generated descriptions, suggesting better alignment with human judgments about the underlying time series.
>
> We also incorporate several mechanisms to mitigate template rigidity and enhance diversity.
> - For each attribute, we design multiple alternative templates and randomly sample among them during caption generation, yielding diverse yet semantically consistent sentences.
> - We randomly shuffle the order of different attribute descriptions so that the final captions do not follow a fixed pattern.
> - Our pipeline is compatible with optional LLM-based refinement, allowing a large language model to restructure the templated sentences while preserving their attribute semantics.
>
> We have now added illustrative examples of generated captions in Appendix B.4 and clarified how this design balances cross-domain robustness with expressive and interpretable descriptions.
>
> > **W4:** GENTS is noticeably slower than non-diffusion baselines.
>
> We thank the reviewer for highlighting the efficiency gap between GenTS and non-diffusion baselines.
>
> We believe that the efficiency of GenTS not only surpasses other diffusion-based baselines but also foundation-model baselines, which is acceptable for most real-world scenarios. As shown in Tab. 9 of our paper, we compare the inference time of GenTS with various baselines on the real-world Weather dataset, with all experiments conducted on a single A40 GPU.
> - GenTS achieves lower inference time than both diffusion-based and foundation-model baselines.
> - The absolute inference time is under one second, which is acceptable for many offline or low-frequency real-world forecasting applications.
>
> Although diffusion models are inherently less efficient due to their multi-step sampling, several acceleration strategies can be applied.
> - In our implementation, we utilize DDIM [1], which introduces a deterministic, non-Markovian diffusion process that significantly speeds up inference compared to standard diffusion sampling.
> - Techniques such as knowledge distillation [2] or latent-space diffusion [3] could further improve efficiency. While our primary focus is on achieving high-quality, unified modeling, improving inference speed remains a promising direction for future work.
>
> [1] Denoising Diffusion Implicit Models. ICLR 2021.
>
> [2] Progressive Distillation for Fast Sampling of Diffusion Models. ICLR 2022.
>
> [3] High-Resolution Image Synthesis With Latent Diffusion Models. CVPR 2022.
>
> > **W5:** A uniform setup is used across datasets (700 epochs, bs=1024, lr=1e-3) with 3 seeds on an A40; fairness to all baselines is unclear.
>
> We thank the reviewer for the careful comment on the training setup and the fairness of our comparisons.
>
> We choose the uniform hyperparameter setting for GenTS across different datasets to demonstrate its *robustness and low tuning cost across diverse domains*. The choice of training GenTS with a fixed setup on all datasets is deliberate. We adopt a single global configuration, which reduces tuning cost and shows that GenTS can perform well without dataset-specific hyperparameter search.
>
> We have taken concrete steps to ensure *a fair comparison to all baselines*.
> - All experiments were run on the same GPU within the same software environment.
> - Every method is trained and evaluated with 3 different random seeds, and we report the mean and standard deviation over these runs.
> - We reproduced baselines using the official implementation and followed the recommended hyperparameter configurations from the original papers.
>
> We promise to open source our reproducible codes including framework, training and evaluation pipeline, datasets upon the acceptance of our paper.

---

> ### Author Response · Authors · 2025-11-24
> **Rebuttal by Authors (part 3 of 3)**
>
> > **W6:** In Table 2 (Weather), J-FTSD shows a large scale gap (GENTS = 1.320 vs. ~290 for two strong baselines); please further explain this anomalous advantage.
>
> We thank the reviewer for highlighting the large performance gap on the Weather dataset under the J-FTSD metric. We would like to clarify that this substantial performance difference is not an anomaly, but rather a direct consequence of our model's design.
>
> The large improvement in J-FTSD reflects GenTS's strength in jointly modeling time series and unstructured textual descriptions, unlike baselines that rely only on structured metadata.
> - GenTS learns to generate time series that closely align with the semantics expressed in the conditioning text, rather than matching coarse metadata. This naturally gives GenTS an advantage over baselines that can only model limited structured information.
> - The J-FTSD metric is designed to measure the similarity of joint distributions, which GenTS models more effectively. As detailed in Appendix F.1 of our paper, J-FTSD embeds both time series and textual descriptions into a shared embedding space, and computes the Fréchet distance between the real and generated joint distributions. GenTS's strong capability in multimodal joint modeling therefore leads to the large performance improvement.
>
> J-FTSD computes the squared Euclidean distance between combined embeddings from two modalities. Its numerical fluctuation range is influenced by the geometry of the multimodal encoder’s embedding space. Performing distance calculations in a normalized embedding space can reduce the absolute scale of J-FTSD across different methods, but it does not alter their relative ranking.
>
> > **W7:** Beyond automatic metrics, consider adding a small human preference study (fidelity and instruction alignment) to support the conclusions.
>
> We thank the reviewer for the helpful suggestion on incorporating a human preference study. We fully agree that human evaluation is important, and we have already adopted such evaluation when assessing the quality of our generated domain-agnostic captions, as detailed in Appendix B.3 of our paper.
>
> We would first like to clarify that the automatic metrics already provide strong evidence regarding fidelity and instruction alignment.
> - Fidelity. The J-FTSD metric was originally proposed in TimeWeaver [1] and is designed to measure the distributional similarity between generated and real data pairs, directly reflecting generation fidelity. Details are provided in Appendix F.1.
> - Instruction alignment. The CTTP score quantifies the semantic alignment between a generated time series and its textual instruction. CTTP is trained via contrastive learning using multiple pairs of time-series and text , as described in Appendix F.2.
>
> To directly address the reviewer's concern, we additionally conduct a small-scale human study on the Weather dataset, as its time series exhibit patterns that are relatively easy for humans to identify.
> - Experiment setting. We compare 20 outputs of three models, GenTS, TimeWeaver [1], and TEdit [2], under the same textual instructions. For each sample, three participants are asked to rank the generated series from 1 (best) to 3 (worst) based on fidelity and instruction alignment. We report the mean and standard deviation of ranks for each model.
> - Experiment results. As shown in Table 2, the ranking results for instruction alignment are consistent with the automatic metrics: GenTS is preferred over TimeWeaver and TEdit. However, we observe that participants struggle with fidelity judgments, resulting in large standard deviations across subjects. This suggests that humans may find it difficult to evaluate time series fidelity, and thus the applicability of human evaluation for time series generation should be considered with caution.
>
> #### Table2. Human preference study of time series generation on Weather dataset. $\downarrow$ means the lower the better.
> |Model|Fid rank $\downarrow$|Instr rank $\downarrow$|
> |:--:|:--:|:--:|
> |TimeWeaver|$2.033_{\pm 0.184}$|$2.100_{\pm 0.082}$|
> |TEdit|$2.000_{\pm 0.308}$|$2.267_{\pm 0.165}$|
> |GenTS (ours)|$1.967_{\pm 0.409}$|$1.633_{\pm 0.155}$|
>
> [1] Time Weaver: A Conditional Time Series Generation Model. ICML 2024.
>
> [2] Towards Editing Time Series. NeurIPS 2024.

---

### Author Response · Authors · 2025-11-24
**General Responses**

We thank all the reviewers for their insightful comments and constructive questions. They have helped us refine our work and better clarify our motivations and contributions for a broader readership.

We are encouraged by the positive feedback, including that:
- The motivation behind our work is both novel and valuable, and it introduces a unified framework that generalizes across multiple time-series tasks (Reviewers 1vhp, 7hBc, woXw).
- The model achieves superior performance across multiple disparate time-series tasks compared with several competitive and recently proposed foundation and diffusion baselines (Reviewers 1vhp, 6ZFC, 7hBc, woXw).
- The paper is comprehensive, clear, and easy to follow (Reviewers 1vhp, 6ZFC, 7hBc).
- The paper introduces a domain-agnostic captioning method to mitigate multimodal data scarcity (Reviewer woXw).

The reviewers' comments have been very helpful in improving the quality of this work. In response, we have made the following revisions (highlighted in blue in the revised paper):
- Clarified the definition of unified modeling and our dataset-specific training setup for GenTS, as suggested by reviewers 6ZFC and 7hBc.
- Added stronger baselines including recent time-series foundation models and conducted a human preference study on the Weather dataset, as suggested by reviewer 1vhp.
- Performed ablation studies on cross attention and diffusion-step embeddings, as suggested by reviewers 6ZFC and woXw.
- Enriched qualitative visualizations of time series editing and text generation results, including more editing examples and representative generated captions, as suggested by reviewers 1vhp and 6ZFC.

---

### Comment · Area_Chair_4Hz7 · 2025-11-28
**Official Comment by Area Chair**

Dear Reviewers,

The discussion phase will end soon. Please take a moment to read the authors’ responses carefully and actively engage in the discussion with the authors and your fellow reviewers.

Thanks for your efforts and contributions to ICLR 2026.

Best regards,

Your AC

---

### Meta-Review · Area_Chair_93Kt · 2026-01-07

**Summary:**

This paper proposes a clear unified diffusion-style framework (GenTS) for multimodal time series modeling with text, and the empirical evaluation across forecasting, generation, and editing is broad and generally well presented. The main concerns focused on (i) novelty/framing, where multiple reviewers felt the core ideas overlap with prior conditional generation and editing work and questioned whether “unified modeling” is an accurate characterization given the training setup; (ii) clarity and fairness of the evaluation setting, especially whether GenTS is trained per dataset/task versus a foundation-style pretraining setup, and whether baseline tuning and comparisons are fully matched; and (iii) practicality and mechanism, including diffusion efficiency, the rigidity/limits of tsfresh-template captions, and limited mechanistic justification beyond additional ablations.

**Reviewer Concerns:**

The rebuttal addressed clarity issues (training paradigm, editing protocol), strengthened baselines (including newer zero-shot forecasters), and added ablations and a small human study.

**Reviewer Scores:**

Reviewer 1vhp and woXw might have moved slightly upward, given the added baselines/ablations and clarified setup. Reviewer 6ZFC and 7hBc might have increased marginally after the clarification.

---

### Decision · Program_Chairs · 2026-01-26

Reject